# *Myo9b* mutations are associated with altered dendritic cell functions and increased susceptibility to autoimmune diabetes onset

Jing Zhang[1,9], Yuan Zou[1,9], Longmin Chen[1,2,9], Fei Sun[1], Qianqian Xu[1], Qing Zhou [1], Yi Wang [1], Xi Luo[1], Na Wang[1], Yang Li[1], Shu Zhang[1], Fei Xiong [1], Ping Yang[1], Shiwei Liu[3], Tao Yang [4], Jianping Weng [5], Décio L. Eizirik[6], Jinhua Yan [7] ✉, Zhiguang Zhou[8] ✉ & Cong-Yi Wang [1] ✉

The regulation of autoimmunity against pancreatic islet β cells for type 1 diabetes (T1D) onset is still unclear. NOD/ShiLtJ (NOD) mice are prone to the onset of autoimmune diabetes, but its congenic strain, ALR/Lt (ALR), is not. Here we show that dendritic cells (DC) in ALR mice have impaired migratory and T-cell priming capability. Genomic comparative analysis maps a 33-bp deletion in the ALR *Myosin IXb* (*Myo9b*) gene when compared with NOD genome; meanwhile, data from knock-in models show that this ALR *Myo9b* allele impairs phenotypic and functional maturation of DCs, and prevents the development and progression of spontaneous autoimmune diabetes in NOD mice. In parallel, while the ALR 33-bp deletion of *Myo9b* is not conserved in human, we find a *MYO9B* R133Q polymorphism associating with increased risk of T1D and enhanced DC function in patients with T1D. Our results thus hint that alterations in Myo9b may contribute to altered DC function and autoimmune diabetes onset.

Type 1 diabetes (T1D) is an autoimmune disease characterized by the severe destruction of insulin-secreting β cells. It has a complex etiology, in which genetic and environmental factors are involved[1], leading to a deleterious "dialogue" between the immune system and the target β cells[2]. The initial stage of T1D pathogenesis is probably triggered by the islet resident antigen-presenting cells (APCs), primarily dendritic cells (DCs), which pick up self-antigens released by the stressed or injured β cells and transport them into the draining lymph nodes (dLNs) to prime the autoreactive T cells. Upon activation, these autoreactive T cells migrate into the islets, where they mediate β cell killing

and exacerbate autoimmunity[3]. In contrast to the well-known role of immunogenic DCs in priming T cells, DCs have been also shown to induce tolerance, depending on their metabolism, maturation/activation state, the molecules expressed on their surface, and cytokines produced[4]. Mature DCs usually stimulate and activate immune response, while immature DCs induce immune tolerance through induction of either T-cell anergy and regulatory T cells (Tregs), or production of regulatory cytokines[5,6]. DCs also undergo a profound metabolic reprogramming based on their activation status. Acquisition of an immunogenic phenotype by DCs, characterized by the enhanced

[1]Department of Respiratory and Critical Care Medicine, the Center for Biomedical Research, NHC Key Laboratory of Respiratory Diseases, Tongji Hospital Research Building, Tongji Hospital, Tongji Medical College, Huazhong University of Science and Technology, Wuhan, China. [2]Department of Rheumatology and Immunology, the Central Hospital of Wuhan, Tongji Medical College, Huazhong University of Science and Technology, Wuhan, China. [3]Shanxi Bethune Hospital, Shanxi Academy of Medical Sciences, Tongji Shanxi Hospital, Third Hospital of Shanxi Medical University, Taiyuan, China. [4]Department of Endocrinology, the First Affiliated Hospital of Nanjing Medical University, Nanjing, China. [5]Department of Endocrinology, the First Affiliated Hospital, Division of Life Sciences and Medicine, University of Science and Technology of China, Hefei, China. [6]ULB Center for Diabetes Research, Université Libre de Bruxelles, Brussels, Belgium. [7]Department of Endocrinology and Metabolism, Guangdong Provincial Key Laboratory of Diabetology, the Third Affiliated Hospital of Sun Yat-sen University, Guangzhou, China. [8]Diabetes Center, the Second Xiangya Hospital, Institute of Metabolism and Endocrinology, Central South University, Changsha, China. [9]These authors contributed equally: Jing Zhang, Yuan Zou, Longmin Chen. ✉e-mail: yanjh79@163.com; zhouzhiguang@csu.edu.cn; wangcy@tjh.tjmu.edu.cn

migratory and overall T-cell priming capacity, is accompanied by, and dependent on a switch from oxidative phosphorylation to glycolysis[7]. Although many non-HLA susceptible genes or loci of T1D have been identified to affect T-cell function and β cell apoptosis[2], less focus has been given to the genes that affect DCs. Therefore, identifying susceptible genes associated with DC function is critical for better understanding the mechanisms underlying T1D pathogenesis and may have important clinical implications.

The NOD/ShiLtJ (NOD) mouse is a well-established model for elucidating T1D etiopathogenesis, as it has genetic and immunological similarities to human disease[8]. The ALR/Lt (ALR) mouse arises from the same outbred Swiss progenitors as the NOD strain, and it shares approximately 85% of the NOD genome[9]. However, unlike NOD mice, ALR mice never develop autoimmune diabetes. Previous studies including our own, have revealed that pancreatic β cells from ALR mice have extraordinary resistance to oxidative stress[10]. Additionally, ALR mice are fully protected from the transfer of diabetogenic CD4[+] BDC2.5 T-cell clone, as well as CD8[+] CTL clones (AI4 and G9C8)[11,12]. These results suggest that the unique 15% ALR likely encodes protective loci against autoimmune diabetes. Indeed, genetic studies revealed that ALR-derived protective loci are linked to Chr. 3, 8, and 17[13] as well as to *mt-Nd2[a]* encoded by the mitochondrial genome (mtDNA)[14]. The protective locus on Chr. 3 overlaps with the *Suppressor of Superoxide Production* (*Susp*) locus, and the Chr. 17-derived marker associated with maximum protection is proximal to *H2-K* end of the *MHC* gene[13]. *Insulin-dependent diabetes* (*Idd*)22 locus is unique because it is the only T1D-resistance linkage that has mapped to mouse Chr. 8, but the nature of the genes involved remain to be elucidated. In the current study, we identified a 33-bp deletion within the ALR *Myo9b* gene on *Idd*22 as compared to that of NOD genome.

Myosin IXb (Myo9b) is a Rho GTPase-activating protein (RhoGAP), which functions as an actin-based molecular motor. It promotes GTP hydrolysis of the small GTPase RhoA and thus converts it from an active GTP-bound state to an inactive GDP-bound state[15]. Myo9b is abundantly expressed in various cell types of the immune system and contributes to the regulation of their morphology and motility[16–18]. Genetic variations in *MYO9B* are associated with inflammatory bowel disease (IBD), Crohn's disease, and ulcerative colitis (UC) in certain populations[19–21], and a role of this gene in increasing susceptibility to systemic lupus erythematosus (SLE) and rheumatoid arthritis (RA) has also been reported, suggesting that polymorphisms at this chromosomal locus might contribute to different autoimmune diseases[22]. However, the association of *MYO9B* with T1D risk remains controversial[23,24].

Herein we report that ALR-derived *Myo9b* manifests a 33-bp deletion, by which it prevents the initiation and progression of spontaneous autoimmune diabetes by regulating phenotypic, metabolic, and functional maturation of DCs. ALR *Myo9b* knock-in (KI) DCs are featured by the increased levels of RhoA and PTEN activity along with downregulated Akt signaling and glycolytic metabolism. We also identify a novel variant in *MYO9B*, which is associated with a higher T1D risk in humans. Collectively, our findings hint the association between *Myo9b* and T1D susceptibility and provide a potential explanation for the underlying mechanisms by linking a potential candidate gene to key metabolic changes in DCs.

## Results

### ALR-derived DCs manifest impaired motility and T-cell priming coupled with a 33-bp deletion in the *Myo9b* gene

Given the importance of DCs in initiating adaptive immune responses in T1D, we first compared their motility and T-cell priming capability between autoimmune diabetes-prone NOD and diabetes-resistant ALR mice. We cultivated stimulated NOD and ALR BMDCs in three-dimensional collagen gel matrices and monitored their spontaneous

migration under a time-lapse microscopy (Fig. 1a). ALR BMDCs showed truncated migratory paths, while no perceptible difference regarding the directionality was detected (Fig. 1b). Similarly, following 24 h skin painting with FITC, the proportions of FITC[+] DCs in the dLNs were lower in ALR mice (Fig. 1c), indicating a decreased migration toward dLNs in vivo. We next analyzed the function of DCs to promote T-cell proliferation in vitro, and found that ALR BMDCs had an impaired ability to initiate the proliferation of CD4[+] BDC2.5 T cells (Fig. 1d).

To identify the genetic foundation underlying above phenotypical differences, we performed high throughput genomic sequencing and subsequent comparative analysis between NOD and ALR mice. Sixty-six InDels among 80,000 genetic variants were mapped to Chr. 8, but only 3 of which led to a sequence great shift (exonic InDels that are not integral multiples of 3 or more than 12 bp) (Fig. 1e). In particular, *Myo9b* located in the *Idd*22 locus was characterized by a 33-bp deletion in the ALR genome (Fig. 1f and Supplementary Fig. 1a). Previous studies revealed that Myo9b is an important signaling component required for the migration of DCs by controlling Cofilin and myosin II activities via the Rho/ROCK/LIM domain kinase (LIMK) signaling axis (schematically illustrated in Fig. 1g)[16,17]. Intriguingly, although ALR BMDCs did not show a significant difference for Myo9b protein levels (Fig. 1h), the active RhoA (RhoA-GTP) levels, however, were much higher as compared to the NOD counterparts (Fig. 1i). In line with these results, ALR DCs displayed increased levels for the phosphorylated LIMK and myosin light chain (MLC) and the inactive phosphorylated form of Cofilin as compared to NOD DCs (Fig. 1j). Additionally, we did not see a compensatory increase of *Myo9a* expression, the paralog of *Myo9b* (Supplementary Fig. 1b). Taken together, those data suggest that the 33-bp deletion likely impairs Myo9b function, which renders ALR DCs with lower immunogenic capability and overactive Rho signaling.

### ALR *Myo9b* KI and *Myo9b* deficiency prevents autoimmune diabetes development in NOD mice

To dissect whether this 33-bp deletion in ALR *Myo9b* confers genetic protection against autoimmune diabetes, we generated ALR *Myo9b* knock-in NOD (defined as KI thereafter) mice by CRISPR-Cas9-mediated genome editing (Fig. 2a), the 33-bp deletion of which were genotyped by PCR analysis (Fig. 2b). Owing to the unclear role of Myo9b itself in T1D pathogenesis, we also constructed a DC-specific *Myo9b*-knockout mouse model in the NOD background (Fig. 2c). *Myo9b* depletion was validated by genotyping of the tail DNA for the presence of Cre (Fig. 2d) and the flox alleles (Fig. 2e). Moreover, western blot analysis confirmed that Myo9b expression was diminished in BMDCs from the *Itgax*-Cre *Myo9b*[f/f] (referred as KO) mice relative to their wild-type (WT) *Myo9b*[f/f] littermates (Fig. 2f), but was unaffected in CD4[+] T cells (Supplementary Fig. 2a).

The above generated KO and KI mice along with WT littermates were subjected to monitoring of autoimmune diabetes onset. The incidence of diabetes was reduced in both KO and KI mice (Fig. 2g), along with a delayed autoimmune diabetes onset (Fig. 2h). The severity of insulitis in KO and KI mice was also much lower than the WT controls examined at 8-, 12-, and 26-week old (Fig. 2i, j). Decreased islet infiltration was further confirmed by CD11c (Fig. 2k) and CD3 (Fig. 2l) staining. Since NOD mice also spontaneously develop autoimmune comorbidities other than autoimmune diabetes, we therefore examined the impact of Myo9b on the development of systemic inflammation and autoimmunity in 10- to 12-week-old prediabetic mice. The KO and KI mice exhibited decreased infiltration of T lymphocytes in the salivary gland (Supplementary Fig. 2b) and colon (Supplementary Fig. 2c), but no significant difference was noted in the lung, liver, kidney, and heart (Supplementary Fig. 2d–g). Collectively, these results suggest that ALR *Myo9b* KI provides protection for NOD mice against autoimmune diabetes progression and lymphoid infiltration in the

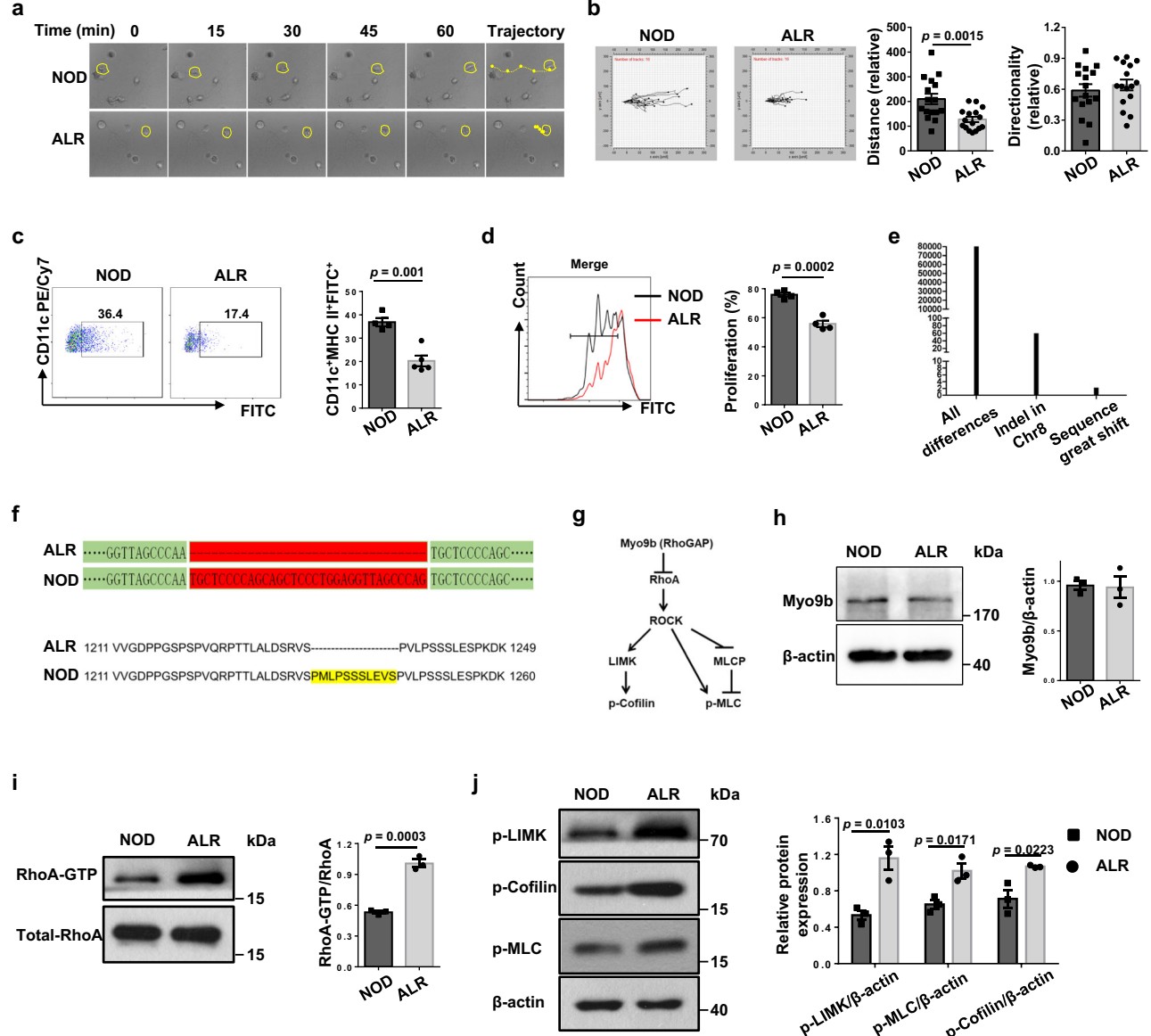

**Fig. 1 | DCs from ALR mice show reduced motility and T-cell priming capability as well as a 33-bp deletion in the *Myo9b* gene. a** Representative migratory paths of NOD and ALR BMDCs in three-dimensional collagen gel matrices recorded by time-lapse microscopy. **b** Left: the migration tracks of NOD and ALR BMDCs in collagen gels were plotted after normalizing the start point to $x = 0$ and $y = 0$. The $x$ axis represents the direction toward the source of chemoattractant. $n = 16$ cells for each group. Right: Quantification of migration distance and directionality of DCs. **c** The percentages of FITC+ DCs in dLNs of NOD ($n = 4$) and ALR ($n = 5$) mice at 24 h following FITC skin painting. **d** Proliferation of CFSE-labeled BDC2.5 naive CD4+ T cells incubated with NOD or ALR BMDCs pulsed with BDC2.5 mimotope.

**e** Number of genetic variants in the ALR genome relative to that of NOD mice. **f** Genomic DNA sequencing and amino acid sequence alignment results showed a 33-bp deletion in ALR *Myo9b* gene. **g** Schematic diagram showing canonical Myo9b signaling pathways. **h** Western blot analysis of Myo9b levels in BMDCs from NOD and ALR mice. **i** GST-pull-down and Western blot analysis of RhoA activity in NOD versus ALR BMDCs. **j** Western blot analysis of p-LIMK, p-Cofilin, and p-MLC in BMDCs. Data were collected from four (**d**) or three independent experiments (**h**–**j**). Values are present as mean ± SEM, and unpaired two-sided Student's *t* test was used for data analysis. LIMK LIM domain Kinase, MLC myosin light chain, sequence great shift exonic InDels that are not integral multiples of 3 or >12 bp.

---

pancreatic islets and other tissues, albeit to a slightly lesser extent than that of *Myo9b* deficiency.

### ALR *Myo9b* KI and *Myo9b* deficiency attenuate phenotypic and functional maturation of DCs

The above results prompted us to examine the effect of Myo9b on biological characteristics of DCs. In agreement with the slower auto-immune progression, the KO and KI mice had decreased percentages and numbers of CD11c+MHC II+ DCs in the pancreatic lymph nodes (PLNs) and pancreas (Fig. 3a, b). ALR *Myo9b* KI and *Myo9b*−/− DCs showed decreased expression of MHC class II (Fig. 3c, d), but no significant changes in terms of the expression of MHC class I and

co-stimulatory molecules CD80 and CD86 were detected (Supplementary Fig. 3a, b). Similarly, ALR *Myo9b* KI and KO did not display a perceptible effect on cDC1/ cDC2 distribution (Fig. 3e, f), and both of the subsets had similar expression pattern of MHC and co-stimulatory molecules as total DCs in the PLNs and pancreas (Supplementary Fig. 3c–f). Consistent with previous studies[16,17], *Myo9b*−/− DCs isolated from the pancreatic islets were featured by the spherical morphology with very short dendrites as compared to that of WT DCs (Fig. 3g). Similar morphology was also noted in the KI DCs, but the change was less significant.

We then checked the biological role of Myo9b using BMDCs. The proportions of CD11c+ cells in BMDCs from WT, KO, and KI mice

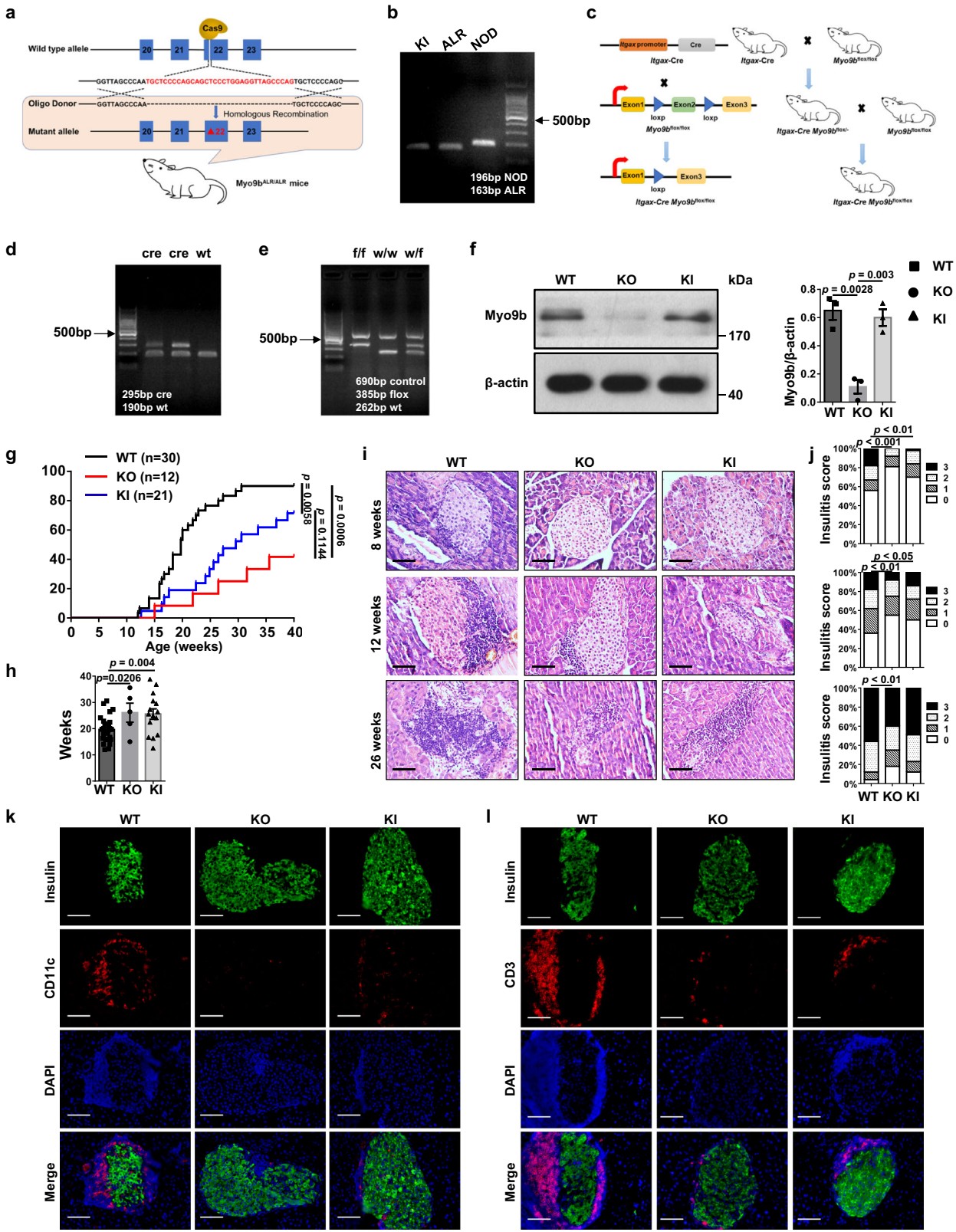

were similar (Fig. 4a), indicating that *Myo9b* KI or KO did not affect DC generation from precursor cells. In response to LPS stimulation in vitro, KO and KI DCs had decreased expression of MHC class II (Fig. 4b), and produced substantially less proinflammatory cytokines IL-6, IL-12, and TNF-α (Fig. 4c). However, no discernable difference was observed in terms of MHC class I, CD80, and CD86 expression (Supplementary Fig. 3g, h), as well as the mRNA levels of

*Ifngr1* and *Il4ra* (Supplementary Fig. 3i). To further dissect the effect of KI and KO on DC migration, transwell assay was carried out. KO and KI DCs had impaired migration ability after LPS stimulation as compared to that of WT DCs (Fig. 4d). In line with these observations, the CFSE-labeled KI and KO DCs displayed decreased migration from periphery toward PLNs than that of WT DCs (Fig. 4e).

**Fig. 2 | ALR *Myo9b* KI and *Myo9b* deficiency prevent the development and progression of spontaneous autoimmune diabetes in NOD mice. a** Strategy for establishing an ALR *Myo9b* knock-in (KI) model in the NOD background by CRISPR-Cas9-mediated genome editing. **b** KI mice were identified by PCR genotyping, with ALR mice as positive control. **c** *Myo9b*^flox/flox mice were generated by inserting two loxP sequences into the intron flanking exon 2 of *Myo9b*, which could produce a nonfunctional Myo9b protein by generating a stop codon in exon 3 after Cre-mediated gene deletion. *Myo9b*^flox/flox mice was then crossed with the *Itgax*-Cre mice to get the DC-specific *Myo9b*-knockout mice. **d, e** Genotyping results of the *Itgax*-Cre (**d**) and flox alleles (**e**). **f** Western blot analysis of Myo9b expression in BMDCs from WT, KO, and KI mice, and a bar graph showing data derived from three mice in

each group. **g** The incidence of diabetes in WT (*n* = 30), KO (*n* = 12), and KI (*n* = 21) mice. **h** The mean age of diabetes onset (WT, *n* = 27; KO, *n* = 5; KI, *n* = 15). **i, j** Representative H&E staining of pancreatic sections (**i**) and insulitis severity scored (**j**) at 8, 12, and 26 weeks of age. **k, l** Representative results for co-immunostaining of insulin with CD11c (**k**) and CD3 (**l**) in pancreas from 12-week-old WT, KO and KI mice. Experiments were repeated three times independently with similar results (**b, d, e**). *n* = 4 per group at each time point (**i–l**). Scale bars: 50 μm (**i, k, l**). Original magnification: ×400 (**i, k, l**). Data are expressed as mean ± SEM. Statistical difference in **g** was analyzed using log-rank test; in **j** was determined by χ2 test; and in other figure parts was analyzed by one-way ANOVA.

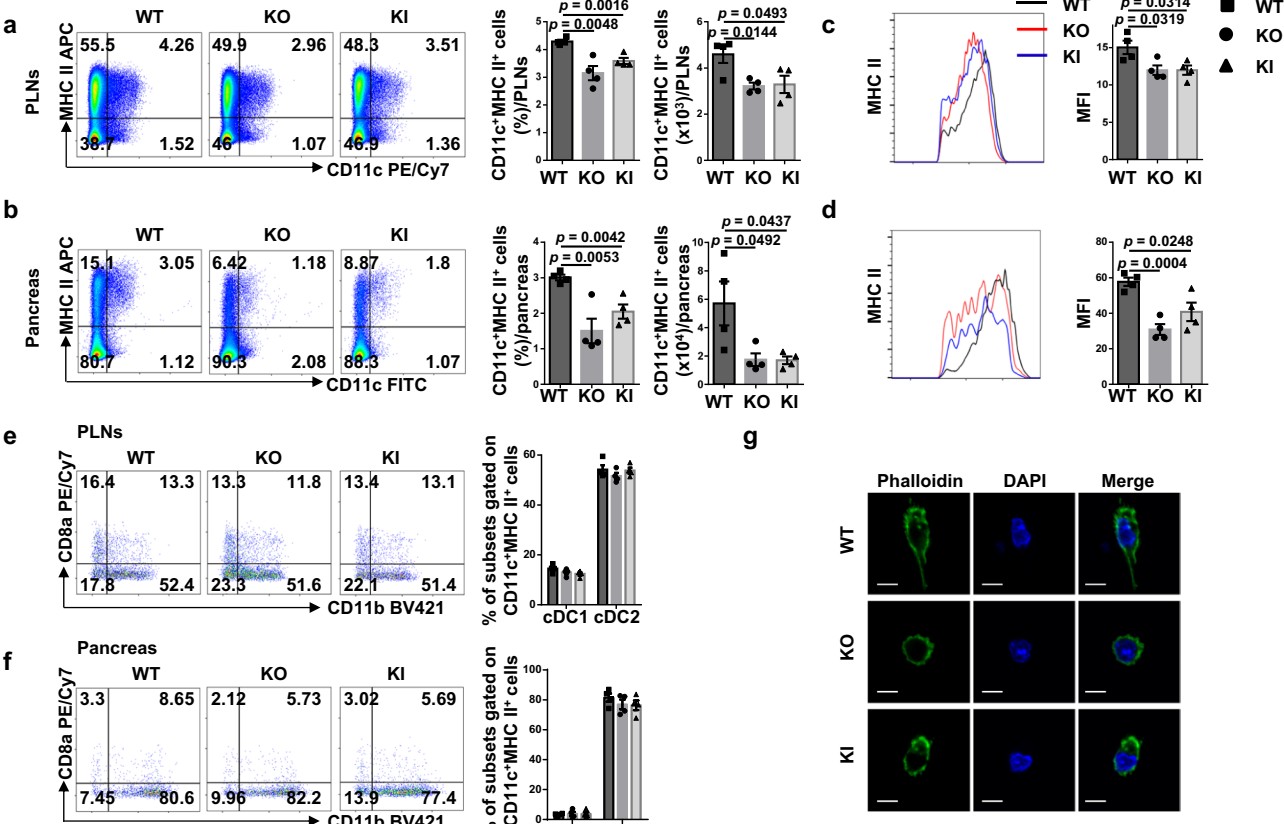

**Fig. 3 | The effect of ALR *Myo9b* KI and *Myo9b* deficiency on DCs in the PLNs and pancreas. a–f** Flow cytometry analysis of PLN and pancreas cells from 10- to 12-week-old WT, KO, and KI mice (*n* = 4 for each group). **a, b** Representative flow cytometry plots, frequencies, and cell numbers of CD11c⁺MHC II⁺ cells in the PLNs (**a**) and pancreas (**b**). **c, d** MHC II expression in CD11c⁺MHC II⁺ cells from the PLNs (**c**) and pancreas (**d**) shown as MFI. **e, f** Representative flow cytometry plots and

frequencies of CD8a⁺CD11b⁻ cDC1 and CD8a⁻CD11b⁺ cDC2 gated on CD11c⁺MHC II⁺ cells in the PLNs (**e**) and pancreas (**f**). **g** Fluorescence images of DCs isolated from pancreatic islets and labeled with FITC-Phalloidin. Scale bars: 10 μm. Values are presented as mean ± SEM. Significance was determined by one-way ANOVA. PLNs pancreatic lymph nodes, MFI mean fluorescence intensity.

Next, we sought to assess the capability of KO and KI DCs to stimulate T-cell proliferation in vitro. It was noted that KO and KI DCs had an impaired function to stimulate the proliferation of BDC2.5 naive CD4⁺ T cells (Fig. 4f). Consistently, adoptive transfer of KO and KI DCs led to a remarkably reduced proliferation of BDC2.5 CD4⁺ T cells in the NOD recipient mice (Fig. 4g), indicating that Myo9b is required for DC function in stimulating T-cell proliferation both in vitro and in vivo. We then checked whether KI and KO affect T-cell polarization program. To this end, we co-cultured LPS-stimulated WT, KO, and KI DCs with BDC2.5 naive CD4⁺ T cells under non-polarized condition. Interestingly, KI and KO DCs displayed reduced capability to induce Th1 cells (Fig. 4h), and Th17 cells but to a less extent (Fig. 4i), while markedly upregulated their capability to induce Treg cells (Fig. 4j). Overall, these data indicate that Myo9b is required for DC mature characteristics, including

MHC class II expression, migration, proinflammatory cytokine secretion, and function in stimulating T-cell proliferation.

## KI and KO DCs suppress T-cell activation and regulate T-cell subpopulations in the PLNs and spleen

Given the crucial role of DCs in T-cell priming and the fact that T1D is characterized by the T cell-mediated autoimmune response against β cells, we first examined T lymphocyte profiling in the PLNs. We failed to detect an observable difference for total CD4⁺ T cells and CD8⁺ T cells between the 10- to 12-week-old nondiabetic KO, KI and WT mice (Fig. 5a). However, the KO and KI mice were featured by the lower percentage of effector T cells along with a higher percentage of naive T cells in total CD4⁺ (Fig. 5b) and CD8⁺ T cells (Fig. 5c). Next, we checked the proportions of T-cell subsets. There was a lower proportion of CD4⁺IFN-γ⁺ Th1 cells (Fig. 5d) and CD8⁺IFN-γ⁺

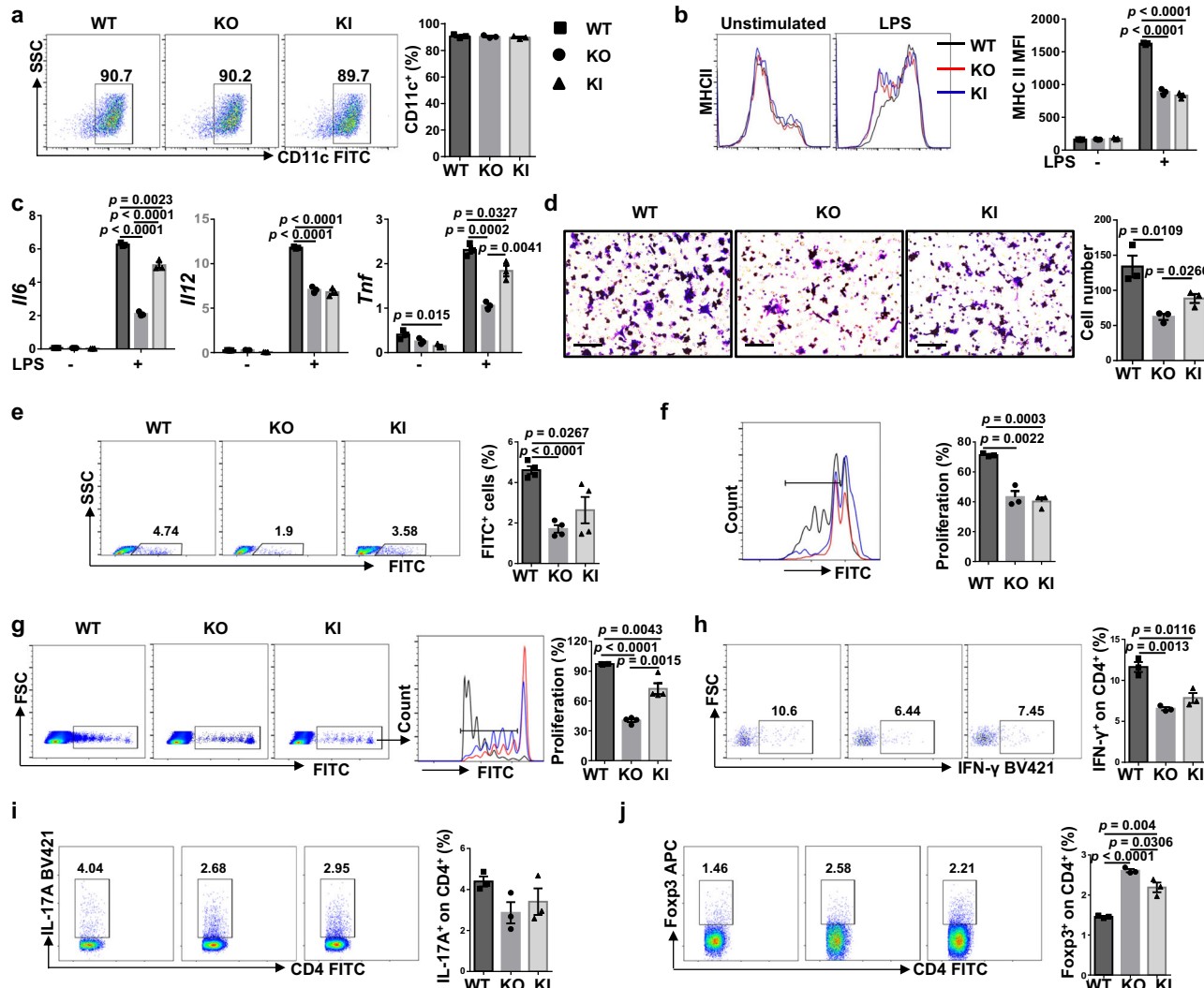

**Fig. 4 | ALR *Myo9b* KI and *Myo9b* deficient DCs have impaired maturation and function. a** The rate of CD11c positive cells in BMDCs of WT, KO and KI mice. **b** Representative histograms and quantitative data of MHC II expression in WT, KO, and KI BMDCs treated with vehicle or LPS for 24 h. **c** RT-PCR analysis of relative mRNA expression of cytokines in BMDCs stimulated with LPS or vehicle for 8 h. **d** Migration of DCs stimulated with LPS for 18 h, analyzed by transwell assay. Scale bars: 100 μm. Original magnification: ×200. **e** In vivo migration of CFSE-labeled BMDCs and quantification results of CFSE⁺ cell proportions in the PLNs within the CD11c⁺MHC II⁺ gate. *n* = 4 per group. **f** Proliferation of CFSE-labeled BDC2.5 naive

CD4⁺ T cells incubated with WT, KO, and KI BMDCs pulsed with BDC2.5 mimotope. **g** In vivo proliferation of CFSE-labeled BDC2.5 naive CD4⁺ T cells in recipient mice (*n* = 4 per group) transferred with WT, KO, and KI DCs, which were pulsed with BDC2.5 mimotope. **h**–**j** Frequencies of Th1 (**h**), Th17 (**i**), and Treg (**j**) cells after co-culture of BDC2.5 naive CD4⁺ T cells with WT, KO, and KI BMDCs pulsed with BDC2.5 mimotope under non-polarized condition. *n* = 3 independent experiments (**a**–**d**, **f**, **h**–**j**). Values are presented as mean ± SEM. Significance was determined by one-way ANOVA.

Tc1 cells (Fig. 5e) in the KO and KI mice, but no significant difference was noted regarding CD4⁺IL-17A⁺ Th17 cells (Fig. 5f). To confirm the above data, we further examined the proportion of Th1 cells in the PLNs in 16-week-old mice. Much higher proportion of Th1 cells was observed (Supplementary Fig. 4a), and the difference between three groups of mice was consistent with that of prediabetic mice. Of note, the proportion and number of CD4⁺Foxp3⁺ Treg cells were significantly higher in KO and KI mice (Fig. 5g and Supplementary Fig. 4b). Similar results were noted in the spleen as well (Fig. 5h–n and Supplementary Fig. 4c). In addition, comparable percentage and number of Treg cells were found in the thymus of WT, KO and KI mice (Supplementary Fig. 4d, e), suggesting that *Myo9b* KI and KO in DCs had little effect on thymic Treg lineage development. However, Treg cells isolated from KO and KI mice harbored stronger suppressive function, as evidenced by the markedly reduced proliferative activity of co-cultured CD4⁺CD25⁻ T cells (Fig. 5o).

Finally, we checked the cytokine profiles in the periphery. The levels of proinflammatory cytokines IL-1β, TNF-α, and IFN-γ were significantly lower, whereas the anti-inflammatory cytokine TGF-β levels were higher in the serum of KO and KI mice (Fig. 5p). Moreover, the KO and KI mice showed increased serum insulin levels (Fig. 5q). Together, our data demonstrate that either *Myo9b* KO or KI attenuate T cell-mediated autoimmune response along with an increased Treg cell proportion and function, thereby inhibiting autoimmune diabetes development.

### *Myo9b* KI confers protection against autoimmune diabetes independent of CD4 T cells

To assess whether the protection conferred by ALR *Myo9b* relies on T cells, we adoptively transferred naive CD4⁺ T cells isolated from 5-week-old BDC2.5 NOD mice into NOD-*scid* (T and B cell deficient) and ALR *Myo9b* KI NOD-*scid* recipient mice, followed by monitoring diabetes incidence (Fig. 6a). As expected, NOD-derived monoclonal CD4⁺

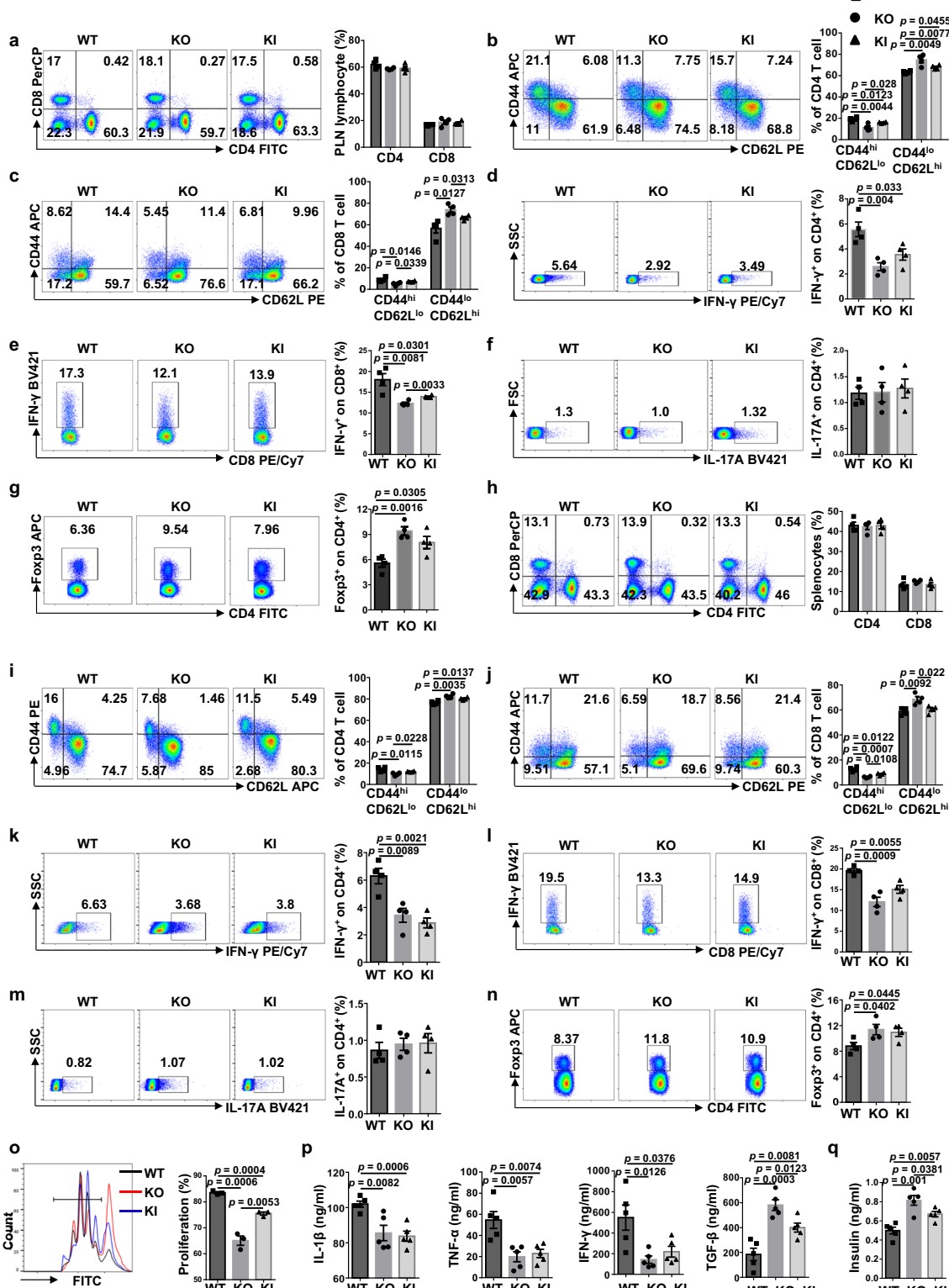

**Fig. 5 | ALR *Myo9b* KI and *Myo9b* deficiency result in decreased prevalence of effector T cells and increased Treg cells in the PLNs and spleen. a–g** PLN cells from 10- to 12-week-old WT, KO and KI mice were harvested and subjected to flow cytometry analysis. Representative FACS plots and frequencies of CD4[+] and CD8[+] T cells (**a**), CD4[+]CD44[high]CD62L[lo] and CD4[+]CD44[lo]CD62L[high] effector/naive T cells (**b**), CD8[+] effector/naive T cells (**c**), CD4[+]IFN-γ[+] (Th1) (**d**), CD8[+]IFN-γ[+] (Tc1) (**e**), CD4[+]IL-17A[+] (Th17) (**f**), and CD4[+]Foxp3[+] (Treg) (**g**) subpopulations are shown. **h–n** Splenocytes from 10- to 12-week-old WT, KO and KI mice were harvested and subject to flow cytometry analysis. Representative FACS plots and frequencies of CD4[+] and CD8[+] T cells (**h**), CD4[+] effector/naive T cells (**i**), CD8[+] effector/naive T cells

(**j**), Th1 (**k**), Tc1 (**l**), Th17 (**m**), and Treg (**n**) subpopulations are shown. **o** Peripheral Treg cells sorted from WT, KO and KI mice were co-cultured with CFSE-labeled WT CD4[+]CD25[−] conventional T cells in the presence of anti-CD3/CD28. CFSE dilution was analyzed by flow cytometry after 72 h, and the percentage of proliferated cells was shown. The experiment was repeated independently three times. **p** Analysis of serum cytokine levels in 12-week-old WT, KO, and KI mice. **q** Serum insulin levels determined in 12-week-old WT, KO and KI mice. *n* = 4 (**a–n**) or 5 (**p, q**) in each study group. Values are expressed as mean ± SEM. Statistical difference was determined by one-way ANOVA.

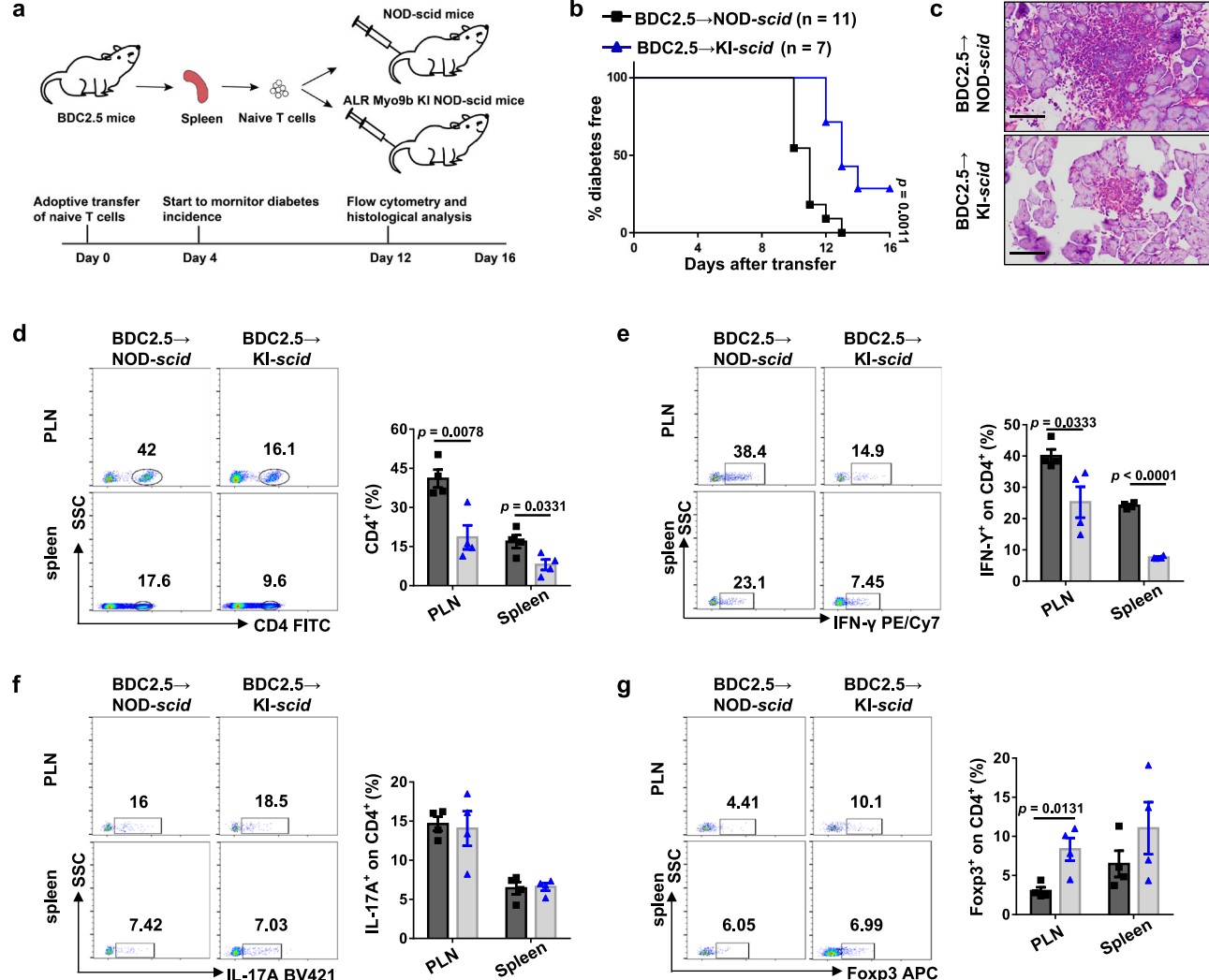

**Fig. 6 | The protective role of ALR *Myo9b* KI is independent of its direct effect on T cells. a** Schematic outline of the experimental design and time line of adoptive transfer model. **b** The incidence of diabetes in ALR *Myo9b* KI NOD-*scid* (n = 7) and NOD-*scid* (n = 11) mice after adoptive transfer of BDC2.5 naive CD4⁺ T cells. **c** Representative results showing H&E staining of pancreases from KI-*scid* and NOD-*scid* mice 12 days after transfer. Scale bars: 50 μm. Original magnification: ×400.

(**d**–**g**) Representative FACS plots and frequencies of CD4⁺ T cells (**d**), Th1 (**e**), Th17 (**f**), and Treg (**g**) subpopulations in the PLNs and spleen are shown. n = 4 per group (**c**–**g**). Data are presented as mean ± SEM. Statistical difference in **b** was analyzed by log-rank test; data from other figure parts were analyzed by unpaired two-sided Student's t test.

T cells transgenically expressing the diabetogenic TCR efficiently transferred autoimmune diabetes to NOD-*scid* mice, but a significantly lower autoimmune diabetes incidence along with a marked delay of disease onset was observed in KI-*scid* recipients (Fig. 6b). Histological analysis on day 12 following transfer revealed that the KI-*scid* recipients had lower severity of islet destruction than that of NOD-*scid* recipients (Fig. 6c). Unlike the unaltered frequencies of total CD4⁺ T cells observed in KO and KI mice (Fig. 5a, h), the KI-*scid* recipients manifested decreased CD4⁺ T cells in the spleen and PLNs (Fig. 6d), which was possibly due to the attenuated proliferation. Furthermore, the KI-*scid* recipients exhibited lower Th1 ratios (Fig. 6e) in the spleen and PLNs, but without a perceptible difference in Th17 cells (Fig. 6f). The proportion of PLN Treg cells was higher in KI-*scid* than NOD-*scid* controls, and similar tendency was observed in the spleen, although the difference did not reach a statistical significance (Fig. 6g). Altogether, our results suggest that ALR *Myo9b* KI protects NOD mice against autoimmune attack independent of CD4 T cells.

Given the crucial role of β cell death in diabetes progression, we then assessed the impact of *Myo9b* KI on β cell viability. We isolated pancreatic islets from 5- to 6-week-old WT and KI mice, and stimulated with proinflammatory cytokines (IL-1β + TNF-α + IFN-γ). Western blot analysis showed that there was no marked difference in cleaved caspase-3 expression between WT and KI-derived islets (Supplementary Fig. 5), indicating that ALR *Myo9b* KI do not affect β cell susceptibility to combination of inflammatory cytokine challenge.

## The KI and KO DCs exhibit defective glycolytic metabolism

To dissect the potential mechanisms by which Myo9b modulates DC characteristics, RNA sequencing was conducted. As compared to WT DCs, the KO and KI DCs displayed 3879 and 2019 differentially expressed genes (DEGs), respectively, and 1165 of them were shared by both sets (Fig. 7a). KEGG analysis of the overlapped DEGs indicated that *Myo9b* KO and KI led to the downregulation of T1D and glycolysis related pathways, among the pathways characterized (Fig. 7b). Specifically, genes involved in glycolytic metabolism were significantly downregulated in KO and KI DCs (Fig. 7c), and we therefore conducted metabolic assays in KI and KO BMDCs. A severely impaired glycolytic

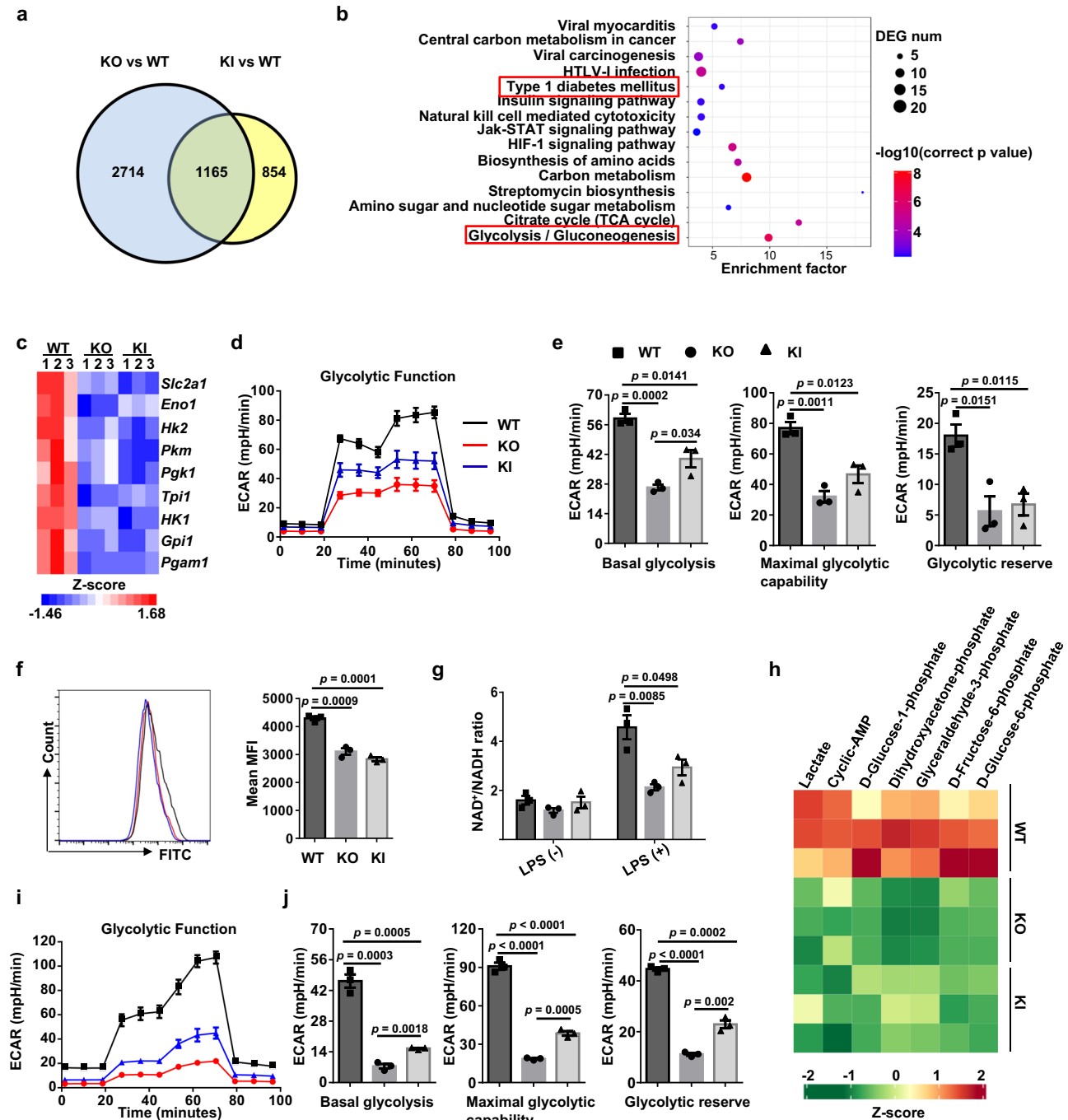

**Fig. 7 | Glycolytic process is inhibited in ALR *Myo9b* KI and *Myo9b*⁻/⁻ DCs. a** Venn diagram showing the numbers of genes altered significantly in BMDCs from KO and KI mice compared to WT mice, as determined by RNA sequencing. **b** Genes downregulated in both KO and KI BMDCs were subjected to KEGG pathway enrichment analysis. **c** Heatmap showing the DEGs relevant to glycolysis in BMDCs. **d** ECAR of BMDCs derived from WT, KO, and KI mice following 24 h of LPS stimulation, which was assessed before and after sequential treatment with glucose, Oligo, and 2-DG. **e** Baseline glycolysis, maximal glycolytic capacity, and glycolytic reserve were shown. **f** Glucose uptake assay carried out to determine the ability of glucose usage by WT, KO, and KI BMDCs. **g** NAD⁺ versus NADH ratio in BMDCs treated with vehicle or LPS for 24 h. **h** Heatmap showing levels of significantly changed glycolytic metabolites in BMDCs upon LPS stimulation for 18 h. **i** ECAR of splenic DCs sorted from WT, KO, and KI mice. **j** Accordingly, baseline glycolysis, maximal glycolytic capacity, and glycolytic reserve were shown. The data in **a–c, h** were derived from three independent biological replicates, and there were two mice for each biological replicate. Data were collected from three independent experiments for (**d–g, i, j**). Values are expressed as mean ± SEM and one-way ANOVA was employed for data analysis.

process was observed, as evidenced by the reduction of ECAR (Fig. 7d) coupled with a marked attenuation of basal glycolysis, maximal glycolytic capacity, and glycolytic reserve (Fig. 7e), while there was no marked difference in the OCR level (Supplementary Fig. 6). The compromised glycolysis in KO and KI BMDCs was further confirmed by the reduction of glucose uptake (Fig. 7f), the ratio of nicotinamide adenine

dinucleotide (NAD⁺) to its reduced form (NADH) (Fig. 7g), and multiple glycolytic intermediate production (Fig. 7h). Consistently, we also found an attenuated glycolytic metabolism in splenic DCs sorted from KO and KI mice (Fig. 7i, j). Collectively, our data indicate that ALR *Myo9b* KI and *Myo9b*⁻/⁻ DCs are characterized by a decreased commitment to glycolysis.

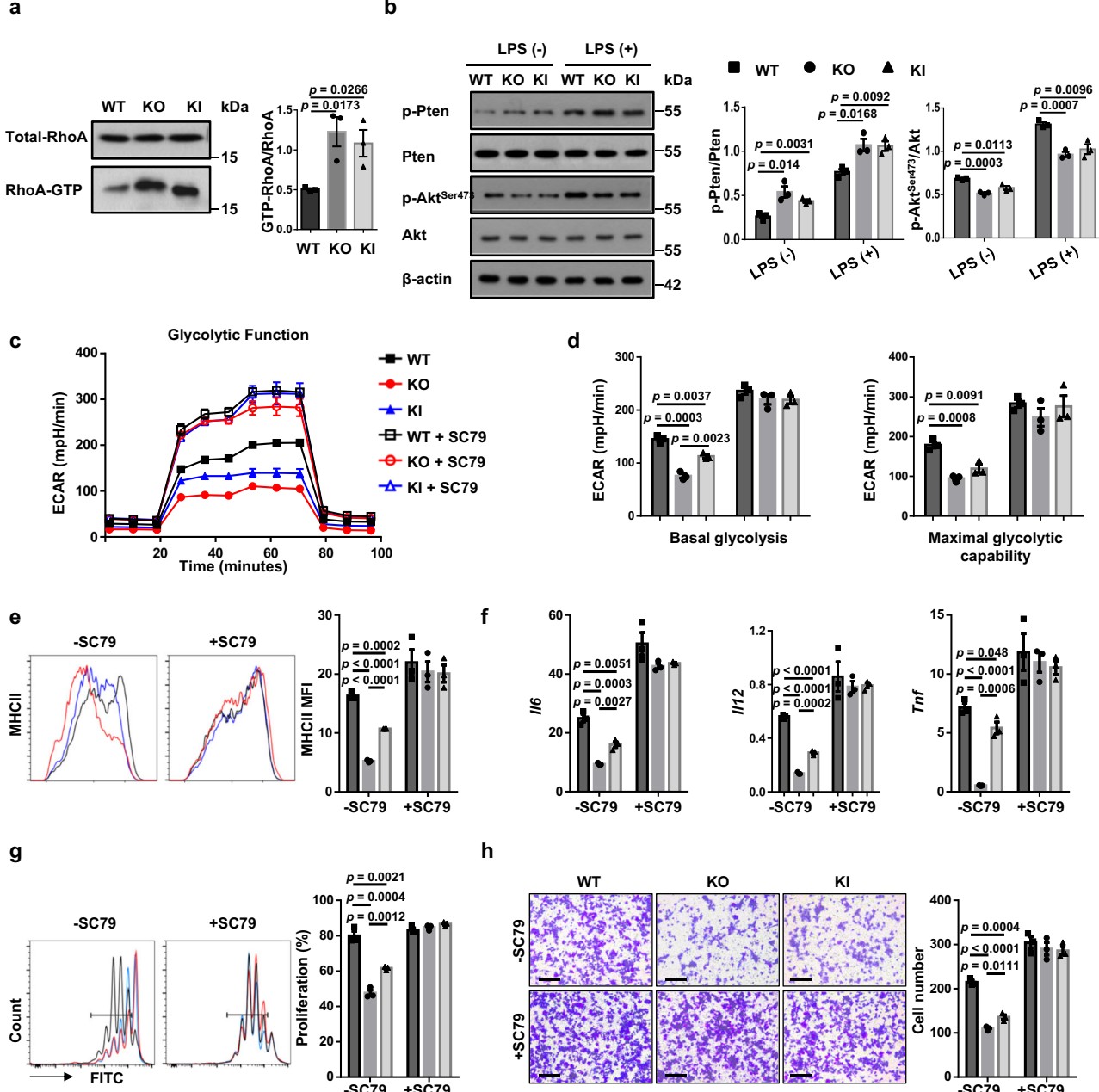

**Fig. 8 | ALR *Myo9b* KI and loss of *Myo9b* upregulate RhoA/Pten signaling and repress Akt-dependent glycolytic process. a** GST-pull-down and western blot analysis of RhoA activity in WT, KO, and KI BMDCs. **b** Results for (p-)PTEN and (p-)AKT expression in DCs with or without LPS stimulation for 1 h. **c** ECAR of WT, KO, and KI DCs pretreated with or without SC79 for 1 h and then stimulated with LPS for 24 h. **d** Baseline glycolysis and maximal glycolytic capacity in BMDCs as shown in **c**. **e** Expression of MHC II of WT, KO, and KI DCs pretreated with or without SC79 for 1 h and then stimulated with LPS for 24 h. **f** RT-PCR analysis of relative mRNA expression of proinflammatory cytokine-encoding genes in DCs pretreated with or without SC79 for 1 h and then stimulated with LPS for 8 h. **g** Proliferation of CFSE-labeled BDC2.5 naive CD4⁺ T cells incubated with SC79- or vehicle-pretreated and BDC2.5 mimotope-pulsed WT, KO, and KI BMDCs. **h** Migration of DCs pretreated with or without SC79 for 1 h and then stimulated with LPS for 18 h, analyzed by transwell assay. Scale bars: 100 µm. Original magnification: ×200. $n = 3$ independent experiments (a-h). Values are presented as mean ± SEM. Statistical significance was determined by one-way ANOVA.

### *Myo9b* KI and KO inhibit Akt-dependent glycolysis by enhancing RhoA/Pten signaling

Akt plays a crucial role in glycolytic process by rapidly providing energy for DC activation and subsequent immunogenic function[25], while active RhoA stimulates the activity of phosphatase and tensin homolog (Pten)[26,27], which is critical to antagonize Akt signaling by dephosphorylating phosphoinositides. We, therefore next, investigated whether KI and KO alter Akt activity through RhoA/PTEN signaling. GST pull-down assays of active GTP-bound Rho revealed that KO and KI BMDCs had increased RhoA-GTP levels (Fig. 8a). Moreover,

an increased Pten phosphorylation was noted in KO and KI DCs, along with markedly reduced levels of phosphorylated Akt (Fig. 8b). Additionally, KO and KI DCs displayed augmented levels of phosphorylated MLC, LIMK, and Cofilin as compared to WT DCs (Supplementary Fig. 7), which were in line with the differences observed between ALR and NOD mice (Fig. 1j).

To address that Myo9b regulates DC characteristics via Akt-dependent glycolysis, an Akt agonist, SC79, was added into DC cultures. Remarkably, SC79 augmented the basal glycolysis and maximal glycolytic capacity in KI and KO DCs to a comparative level as that of

WT DCs (Fig. 8c, d), indicating a crucial role for Akt in *Myo9b* KI and KO-induced impairment of glycolysis. Furthermore, addition of SC79 almost completely abolished the impact of KI and KO on the maturation and function of DCs as shown by the similar levels of MHC II expression (Fig. 8e), proinflammatory cytokine production (Fig. 8f), and function in promoting T-cell proliferation as WT DCs (Fig. 8g). Migratory capability was also in large part restored (Fig. 8h). Taken together, our results demonstrate that *Myo9b* KI and KO disrupt glycolytic process via RhoA/Pten-mediated downregulation of Akt, thereby attenuating DC maturation.

## The rare variant rs764932023 in *MYO9B* is associated with human T1D

Finally, we sought to translate the above findings into T1D setting in humans. Since the 33-bp deletion identified in ALR mice is not conserved between mouse and human, we therefore conducted targeted sequencing of *MYO9B* along with comparative biostatistical analysis in 260 T1D cases and 240 healthy controls (Fig. 9a). The frequencies and annotation of each variant among the cases and controls are summarized in Supplementary Table 1, among which variants Rs764932023, Rs766200985, and Rs776331004 (Fig. 9b) were verified using the Taqman genotyping procedures in a larger cohort that included 1298 T1D cases and 2936 healthy controls. The minor allele frequency (MAF) of variant Rs764932023 differed between the two groups, being more frequent in T1D patients than in controls (OR = 6.39, $p = 1.359 \times 10^{-4}$, 95% CI = 2.169–22.711) (Fig. 9c). Moreover, the results showed that the measured MAF of Rs764932023 was similar to

reported MAFs for ethnicity-matched populations, supporting genotyping accuracy (Supplementary Table 2). The G > A variant of Rs764932023 resulted in an R (Arg) > Q (Gln) amino acid substitution, and sequence analysis indicated that the Arg133 locus of MYO9B is revolutionarily conserved from *Gallus gallus* to *Homo sapiens* (Fig. 9d). Moreover, healthy controls carrying the Rs764932023 polymorphism manifested higher MYO9B protein levels in monocyte-derived dendritic cells (MoDCs) as compared to those without the variant (Fig. 9e). Unfortunately, because of the extreme low frequency of SNPs Rs766200985 and Rs776331004 in the general population, we failed to detect the above association in those two SNPs (Supplementary Fig. 8), although a higher MAF was noted in T1D patients for Rs766200985.

We next characterized the functional relevance of the R133Q polymorphism by assessing its impact on MYO9B-regulated cellular activities. MoDCs derived from 3 T1D $MYO9B^{R133Q}$ carriers and 3 T1D non-carriers with comparative age and disease state were employed for metabolic analysis. Indeed, patients carrying the $MYO9B^{R133Q}$ polymorphism manifested higher glycolytic metabolism (Fig. 10a, b) coupled with increased MHC II expression (Fig. 10c), inflammatory cytokine production (Fig. 10d), and capacity to prime CD4$^+$ T cells (Fig. 10e). To further confirm these results, DCs from *Myo9b* KO mice were transduced with adenoviral particles carrying the vector, $MYO9B^{WT}$, and $MYO9B^{R133Q}$, respectively. The cells were subjected to seahorse metabolic analysis following 36 h of transduction. As expected, transduction of $MYO9B^{R133Q}$ increased MYO9B levels in $Myo9b^{-/-}$ DCs similar to human MoDCs (Supplementary Fig. 9), and $MYO9B^{R133Q}$ significantly increased basal glycolysis and maximal

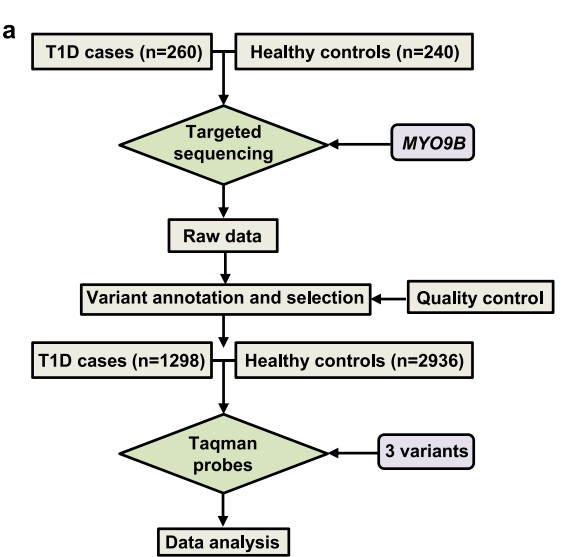

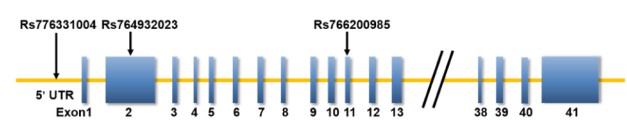

**c**

Rs764932023  G>A  R (Arg) > Q (Gln)

| Subjects | *N* | Genotyping distribution (GG / GA / AA) | MAF |
|---|---|---|---|
| T1D | 1298 | 1284 / 14 / 0 | 0.54% |
| Control | 2936 | 2931 / 5 / 0 | 0.085% |

OR = 6.388, $p = 1.359 \times 10^{-4}$, 95% CI = 2.169-22.711

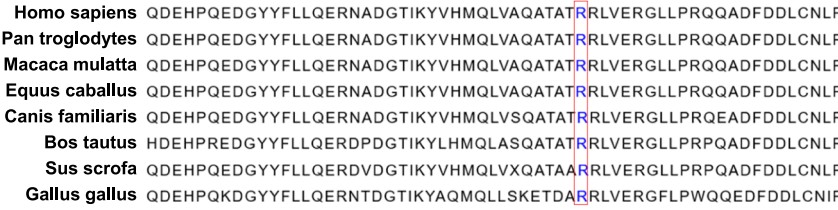

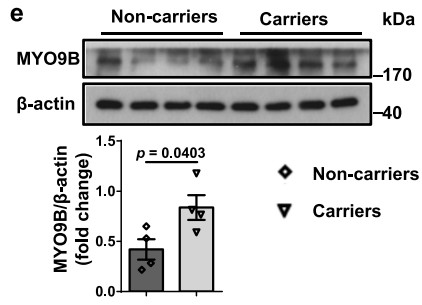

**Fig. 9 | Identification and characterization of the rare variant rs764932023 in *MYO9B*. a** Flow diagram for study design and data analysis. **b** Schematic representation of the approximate locations of variants Rs764932023, Rs766200985, and Rs776331004 relative to the *MYO9B* gene. **c** Genotyping result for Rs764932023 in T1D cases and healthy controls. **d** Alignment of MYO9B sequences in different species. The conserved Arg133 is framed in red. **e** MYO9B expression in MoDCs from healthy controls carrying variant Rs764932023 and those without the variant. *n* = 4 for each genotype. Values are expressed as mean ± SEM. Statistical difference in **c** was assessed using Fisher's exact test; in **e** was determined by unpaired two-sided Student's *t* test.

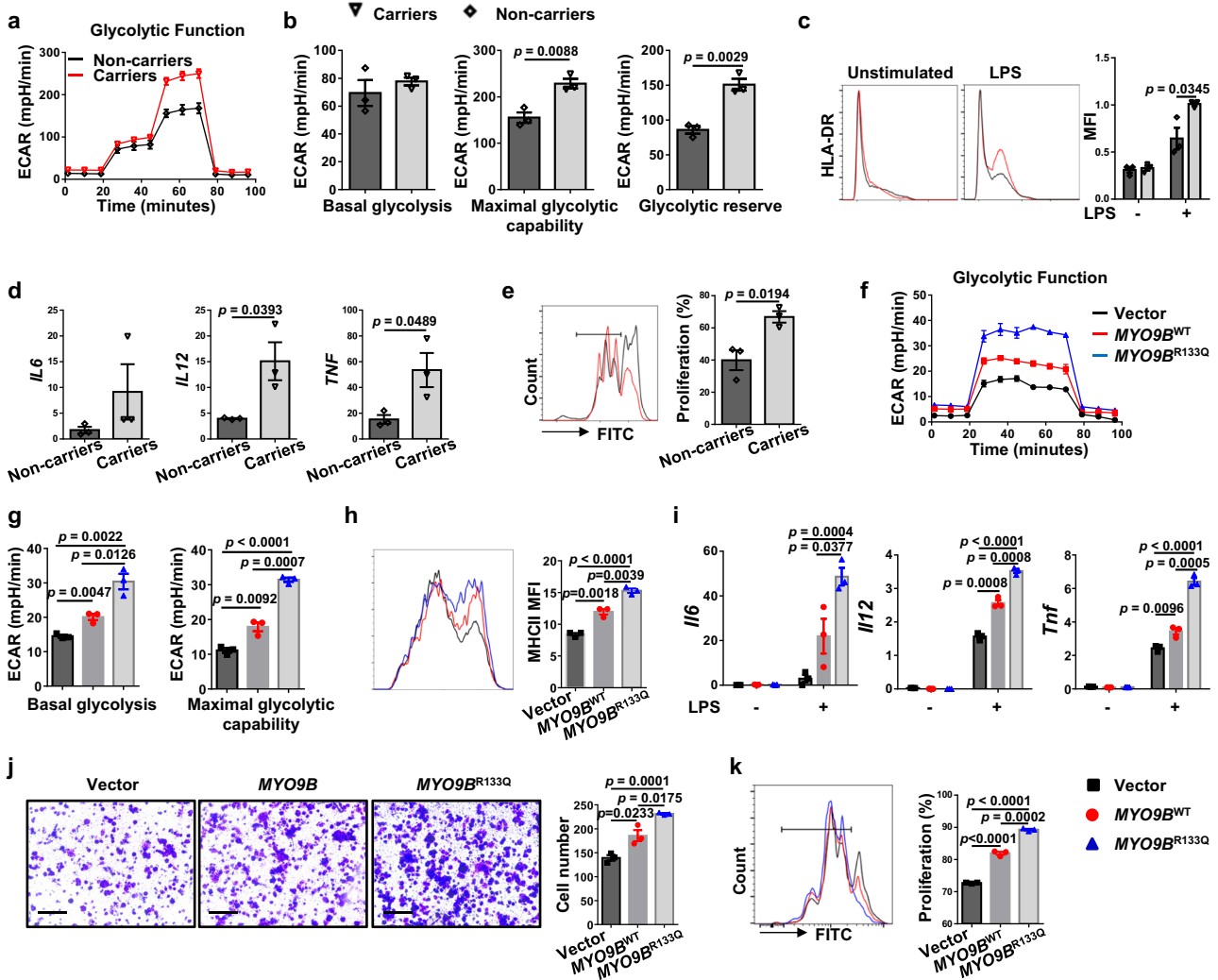

**Fig. 10 | Rs764932023 in *MYO9B* gene is associated with T1D in humans.**
**a**–**e** MoDCs were differentiated from PBMCs of T1D patients carrying the
*MYO9B*[R133Q] polymorphism or not, and subjected to the following experiments. *n* = 3
per genotype. **a** ECAR of MoDCs stimulated with LPS for 24 h. **b** Accordingly,
baseline glycolysis, maximal glycolytic capacity, and glycolytic reserve were shown.
**c** Representative histograms and quantitative data of HLA-DR expression in MoDCs
treated with vehicle or LPS for 24 h. **d** RT-PCR analysis of relative mRNA expression
of cytokines in MoDCs stimulated with LPS for 8 h. **e** Proliferation of CFSE-labeled
allogenic CD4⁺ T cells incubated with MoDCs derived from *MYO9B*[R133Q] carriers and
non-carriers. **f** ECAR of *Myo9b* deficient mouse DCs transduced with adenoviral
particles containing vector, *MYO9B*[WT], or *MYO9B*[R133Q], and then stimulated with LPS
for 24 h. **g** Baseline glycolysis and maximal glycolytic capacity in cells as shown in **f**.
**h** Expression of MHC II of *Myo9b* deficient mouse DCs transduced with vector,

*MYO9B*[WT], or *MYO9B*[R133Q] adenovirus and then stimulated with LPS for 24 h. **i** RT-PCR
analysis of relative mRNA expression of proinflammatory cytokine genes in *Myo9b*
deficient mouse DCs transduced with vector, *MYO9B*[WT], or *MYO9B*[R133Q] adenovirus
and then stimulated with LPS for 8 h. **j** Migration of *Myo9b*⁻/⁻ mouse DCs transduced
with vector, *MYO9B*[WT], or *MYO9B*[R133Q] virus and then stimulated with LPS for 18 h,
analyzed by transwell assay. Scale bars: 100 μm. Original magnification: ×200.
**k** Mouse DCs deficient in *Myo9b* were transduced with vector, *MYO9B*[WT], or
*MYO9B*[R133Q] virus. Shown is the proliferation of CFSE-labeled BDC2.5 naive CD4⁺
T cells incubated with BDC2.5 mimotope-pulsed DCs. Data were collected from
three independent experiments (**f**–**k**). Values are expressed as mean ± SEM. Sta-
tistical difference in **b**–**e** was assessed using unpaired two-sided Student's *t* test; in
**g**–**k** was analyzed by one-way ANOVA.

glycolytic capacity as compared to *MYO9B*[WT], while cells in the vector
group had the lowest level (Fig. 10f, g). Accordingly, KO DCs trans-
duced with *MYO9B*[R133Q] displayed higher levels of MHC II expression
(Fig. 10h), proinflammatory cytokine production (Fig. 10i), migration
(Fig. 10j), and function in promoting T-cell proliferation as compared
to DCs transduced with *MYO9B*[WT] (Fig. 10k). Altogether, these data
support that the Rs764932023 variant correlates with an enhanced DC
function to prime autoreactive CD4⁺ T cells, potentially giving a
mechanistic explanation for its association with increased T1D risk.

## Discussion
T1D is an autoimmune disease caused by the interaction between
genetic alterations and environmental factors, and over 60 T1D sus-
ceptible genes or loci have been identified to date[28]. In this report, we

conducted comparative genomic sequence analysis between NOD (a
human T1D-prone model) and ALR (a NOD congenic strain with auto-
immune diabetes resistance) mice, by which we characterized a 33-bp
deletion in *Myo9b* of ALR mice. Importantly, *Myo9b* is located within
the region of *Idd22* locus, knock-in of ALR *Myo9b* into NOD mice
decreased autoimmune diabetes incidence and insulitis progression,
which was reminiscent of the phenotype observed in our *Itgax*-Cre
*Myo9b*⁻/⁻ NOD mice. CD4 T cells are considered as the major culprits in
the aggravation of insulitis and β cell death in T1D, and it was reported
that *Myo9b*⁻/⁻ CD4 T cells are selectively impaired in crossing the
basement membranes and accumulating in nonlymphoid tissues[18]. To
exclude the potential protective effect brought by ALR *Myo9b* KI CD4
T cells, diabetogenic naive CD4⁺ T cells from BDC2.5 NOD mice were
adoptively transferred into NOD-*scid* mice and ALR *Myo9b* KI NOD-*scid*

recipient mice. KI-*scid* recipients were still more resistance to auto-immune diabetes development, indicating that *Myo9b* KI only manifested a mild impact on CD4 T cells. Similarly, ALR *Myo9b* KI did not show a significant impact on the susceptibility of β cells to the challenge of combination of inflammatory cytokines. However, we cannot completely rule out the feasibility that ALR *Myo9b* impacts the function of additional immune cells other than DCs, which would be a major focus in our future studies.

Myo9b is involved in cell shape and motility, cell-cell junctions and membrane trafficking, via regulation of actin cytoskeleton organization and actomyosin II contractility[15,17]. Consistent with these previous findings, the migratory capacity of KO and KI DCs was reduced both in vivo and in vitro, implying less recruitment of DCs to PLNs after encountering autoantigens in the pancreas. Phenotypical and functional maturation is an essential prerequisite for the immunogenic activity of DCs. Maturation is not only associated with greater ability to migrate to T cell-rich areas, such as PLNs, but also with enhanced antigen processing and presentation through upregulated expression of MHC class II, as well as increased ability to prime naive T cells via elevated co-stimulatory molecule expression and cytokine production[29]. Indeed, we presently observed that ALR *Myo9b* KI and *Myo9b* KO DCs exhibited reduced expression of MHC II, and cytokines IL-6, IL-12 and TNF-α, and decreased ability to stimulate T-cell responses in vitro and in vivo, all of which indicate their impaired maturation and function. Mature DCs usually stimulate immune responses, while immature DCs tend to induce tolerance. Intriguingly, KI and KO DCs showed a higher capacity for Treg cell induction, leading to a consistent increase in the proportion and number of Treg cells in the PLNs. Since CD4+ T cells are recognized as the major culprits due to their enormous influence on the amplification of immune response in T1D setting, this report, therefore, selectively focused on the impact of altered DC maturation on CD4+ T cells. Unexpectedly, flow cytometry analysis also noted lower proportions of CD8+CD44hiCD62Llo effector T cells and IFN-γ+ cytotoxic CD8+ T cells in the PLNs and spleen of KO and KI mice, but absence of altered MHC I expression. This discrepancy could be caused by the functional difference of CD4 T cells, as CD4 T cells also play a critical role in CD8 T-cell activation. However, the effect of ALR *Myo9b* KI and *Myo9b*−/− DCs on CD8+ T cells during the process of cross-presentation, along with a detailed DC classification and contribution would be further investigated in our follow-up studies.

Myo9b is an important signaling component for cell trafficking by controlling Cofilin and myosin II activities via RhoA signaling[17]. We presently detected elevated RhoA-GTP levels in ALR *Myo9b* KI and *Myo9b* deficient DCs, which in turn, increased the inactive phosphorylated form of Cofilin and MLC. However, this could not completely explain the observed phenotype. Recent advances in the field of immunometabolism highlight that DCs markedly alter glycolysis upon activation, which has important implications for their functionality[30]. Pharmacological blockade of glycolysis using 2-DG or genetic deficiency of glycolytic enzymes, such as α enolase (ENO1), prevents BMDC maturation and immunogenicity upon stimulation with LPS or Chlamydia and can skew BMDCs toward inducing Treg cells[25,31,32]. Moreover, mouse splenic DCs decrease IL-12 secretion and the ability to prime CD4+ and CD8+ T cells once activated by LPS in the presence of 2-DG[31], and vitamin D induces a tolerogenic DC phenotype via metabolic re-programing[33]. In fact, KEGG analysis of our RNA-seq data revealed that the glycolytic pathway was inhibited in ALR *Myo9b* KI and *Myo9b* KO DCs, a finding confirmed by seahorse metabolic analysis. The observed reduction in glucose uptake, glycolytic intermediate production, and the ratio of NAD+ to NADH ratio further confirmed that ALR *Myo9b* KI and *Myo9b* KO attenuate glycolysis in DCs. Importantly, the use of a chemical Akt agonist almost completely rescued the glycolytic metabolism, and immature characteristics of KO and KI DCs, suggesting that ALR *Myo9b* KI and

*Myo9b* KO inhibit DC maturation predominantly by hampering Akt-dependent glycolysis.

The association of polymorphisms in *MYO9B* and susceptibility to autoimmune diseases was first found in celiac disease[34], which was further noted in IBD, UC, SLE, and RA[19–22]. Since substantial evidence exists for comorbidity between intestinal disease-related gut permeability and T1D[35,36], two human studies have been performed to evaluate the effects of *MYO9B* gene polymorphisms on T1D patients, focusing on the most studied SNPs Rs962917, Rs2279003, and Rs2305764[23,24]. The results, however, were discrepant. The association was detected in Dutch and Spanish patients, but not in British patients, which might be due to the diverse ethnicity. To explore novel T1D-related *MYO9B* variants in humans, we performed targeted sequencing for *MYO9B* in 260 Chinese T1D cases and 240 healthy controls. Subsequent biostatistical analysis and studies in a larger Chinese cohort confirmed that MAF of variant Rs764932023 of *MYO9B* was highly associated with T1D risk (Odds ratio = 6.388, $p = 1.359 \times 10^{-4}$). To address the functional relevance of this variant on DCs, MoDCs from T1D patients carrying the *MYO9B*R133Q polymorphism and KO DCs transduced with *MYO9B*WT and *MYO9B*R133Q were employed for the studies. The *MYO9B*R133Q variant is associated with increased basal glycolysis and maximal glycolytic capacity along with enhanced capability to induce T-cell activation. Interestingly, the expression levels of MYO9B were increased in MoDCs from healthy controls with the *MYO9B*R133Q variant than those without it, but the detailed mechanism is yet to be addressed. Therefore, our results warrant further investigations by well-designed prospective studies with larger sample size and diverse populations. Furthermore, additional studies allowing stratification for potential confounders ought to be performed to further interpret the potential role of *MYO9B* polymorphisms in T1D etiology. Although this study cannot establish a direct role for *MYO9B* in human T1D, it could point to DC function as a potential research area for investigating the impact of genetic variation on T1D risk.

In conclusion, the present study provides evidence that ALR *Myo9b* KI and loss of *Myo9b* restrains Akt-dependent DC glycolysis via RhoA/PTEN signaling. This metabolic change results in impaired phenotypical and functional maturation of DCs. Consequently, ALR *Myo9b* KI and *Myo9b* deficiency in DCs delay diabetes onset and prevent the progression of autoimmune diabetes in NOD mice. These results established ALR *Myo9b* as a loss-of-function mutation that modulates autoimmune diabetes susceptibility in NOD mice. Human studies further identified the association of a rare variant, rs764932023 (R133Q), in the *MYO9B* gene with higher T1D risk. These findings broaden the current understanding of the role of Myo9b in T1D-related autoimmune initiation, provide additional insights into the genetic basis of T1D risk, and demonstrate a novel crosstalk between the genetic variants and the metabolism/function of DCs. These discoveries may be informative in developing future precision medicine in clinical settings.

## Methods
### Mice
The *Myo9b*flox/flox mice in the C57BL/6 background were generated using the Cre-LoxP system. The *Myo9b*flox/flox NOD mice were generated by backcrossing the *Myo9b*flox/flox C57BL/6 with NOD mice for 15 generations, and their purity of NOD background was verified by DNA sequencing. *Myo9b*flox/flox NOD was then crossed with CD11c-Cre transgenic NOD mice to obtain the DC-specific *Myo9b* deficient NOD mice. ALR *Myo9b* KI NOD and NOD-*scid* mice were generated by the CRISPR-Cas9-mediated genome editing. In brief, single-guide RNAs (sgRNA1: GGTATGGAAGCCGCCTGGTGTGG; sgRNA2: GCTGTGGTACTGACA-GATTGAGG) were selected to control the targeting specificity of the Cas9. Cas9 mRNA, sgRNAs, and donor oligonucleotides were co-injected into zygotes from NOD-*scid* mice, followed by transplantation into pseudopregnant mice. PCR and Sanger sequencing were

performed to select positive F0 mice. Germline transmission was verified by breeding F0 chimeric mice with NOD-*scid* mice and subsequent genotyping and Southern blot analysis of the F1 offspring. ALR mice (003070) and BDC2.5 NOD mice (004460) were originally obtained from Jackson's Laboratory (Bar Harbor, ME, USA), and NOD-*scid* mice (13002 A) were purchased from Beijing HFK Bioscience (Beijing, China). All mice were maintained under specific pathogen-free conditions at the Tongji Hospital Animal Center with 20-24 °C ambient temperature, 45-65% humidity, and a 12/12 h light/dark cycle. After 10 weeks of age, female NOD mice were monitored for blood glucose twice a week using an Accu-Check Advantage glucometer (Roche Diagnostics, Indianapolis, IN, USA) and considered as diabetic once blood glucose levels exceeded 13.8 mmol/l for 3 consecutive days. All animal experiments were performed in accordance with the National Institutes of Health (NIH) guidelines. The study was approved by the Tongji Hospital Animal Care and Use Committee (TJH-201903020).

### Collagen motility assay

BMDCs were mixed with collagen matrix to a final concentration of $3 \times 10^6$ cells/ml, and seeded into the narrow channel (observation area) of an μ-slide chemotaxis chamber (80326; Ibidi, Martinsried, Germany). The narrow channel separates the left and right reservoirs, one of which was supplemented with CCL21 (250-13; PeproTech, Rocky Hill, Connecticut, USA). For the duration of 2–4 h, pictures were taken at 2-min intervals by phase-contrast microscopy. Sixteen randomly selected cells were manually tracked in each assay, and cell migration tracks were analyzed using the ImageJ (NIH) software with the Manual Tracking plug-in and the Chemotaxis Tool plug-in from Ibidi.

### Histological analysis

Pancreatic tissues were fixed with 4% paraformaldehyde, embedded in paraffin, sectioned, and stained with hematoxylin and eosin. Insulitis was scored in a blinded fashion by two pathologists as follows: 0, no infiltration; 1, <25% infiltration of the islet; 2, 25–50% infiltration of the islet; and 3, >50% islet infiltration[37]. For immunostaining, the sections were probed with primary antibodies against insulin (GB12334; servicebio, Wuhan, China), CD11c (GB11059; servicebio) and CD3 (GB13014; servicebio). The antibody dilution was 1/100.

### Transwell assay and FITC-induced DC migration assay

The in vitro migration assay was performed using 24-well Transwell chambers with 8-μm pores (Corning, NY, USA). The RPMI 1640 medium supplemented with 10% fetal bovine serum (FBS, Gibco, Shanghai, China) and 50 ng/ml CCL21 was added into the lower chamber, and BMDCs with a density of $1 \times 10^5$ cells/200 μl were seeded into the upper chamber and incubated for 18 h at 37 °C. In selected experiments DCs were pretreated with vehicle or 4 μg/ml of an Akt agonist, SC79 (HY-18749; MedChemExpress, Shanghai, China), for 1 h. After incubation, the cells were fixed with 4% paraformaldehyde and stained with 0.1% crystal violet. The number of membrane penetrated cells was determined in the randomly selected fields.

For fluorescein isothiocyanate (FITC)-induced DC migration, mice were painted with 200 μl FITC (10 mg/ml) (46950; Sigma-Aldrich Co., St. Louis, MO, USA) dissolved in acetone/dibutyl phthalate (1:1, Sigma-Aldrich Co.) mixture on the shaved abdomen. The number of DCs migrated into the inguinal lymph nodes was analyzed by flow cytometry 24 h later.

### Flow cytometry

For cell surface staining, single-cell suspension was incubated with indicated antibodies for 30 min on ice. Intracellular staining was performed using the Transcription Factor Buffer Set (562574; BD Biosciences, San Diego, CA, USA) with the antibody cocktails. For intracellular cytokine staining, cells were stimulated with Cell

Activation Cocktail (423303; Biolegend, San Diego, CA, USA) for 4-6 h. Antibodies used are listed below: FITC anti-mouse CD11c (117306), PE/Cy7 anti-mouse CD11c (117318), Alexa Fluor 647 anti-mouse I-Ad (115010), Brilliant Violet 421 anti-mouse/human CD11b (101235), Brilliant Violet 421 anti-mouse CD4 (100438), FITC anti-mouse CD4 (100406), PerCP anti-mouse CD8a (100731), PE anti-mouse CD8a (100708), PE/Cy7 anti-mouse CD8a (100722), FITC anti-human HLA-DR (307604), APC anti-mouse CD62L (104412), PE anti-mouse CD62L (104408), APC anti-mouse/human CD44 (103012), PE anti-mouse/human CD44 (103008), Brilliant Violet 421 anti-mouse IFN-γ (505829), PE/Cy7 anti-mouse IFN-γ (505826), Brilliant Violet 421 anti-mouse IL-17A (506926), Alexa Fluor 647 anti-mouse/rat/human FOXP3 (320014), PE/Cy7 anti-mouse CD86 (105014), FITC anti-mouse H-2Kᵈ (116605), and Alexa Fluor 488 anti-DYKDDDDK tag antibody (637317) from Biolegend (San Diego, CA, USA) and PE hamster anti-mouse CD80 (553769) from BD Bioscience (San Diego, CA, USA). The antibody dilution was 1/200. Data were obtained on a MACS Quant Analyzer10 (Miltenyi Biotec, Germany) and analyzed with the FlowJo software (v10.5.3). Gating strategies are shown in Supplementary Figs. 10 and 11.

### Real-time PCR and immunoblot analysis

Total RNA was extracted from cultured cells with TRIzol reagent (Takara, Japan). cDNA synthesis was performed using a HiScript 1st Strand cDNA Synthesis Kit (R111-01; Vazyme, Nanjing, China) according to the manufacturer's instructions. The relative expression of mRNA was quantified by real-time PCR using the Hieff qPCR SYBR Green Master Mix (11203ES03; Yeasen, Shanghai, China) and calculated with the $2^{-\Delta\Delta Ct}$ method as reported[38]. The primer sequences for all examined genes are listed in Supplementary Table 3.

Cells were washed with ice-cold PBS and ruptured with RIPA lysis buffer (Beyotime, Shanghai, China) containing the protease inhibitor cocktail (Roche, Indianapolis, IN). Equalized cell lysates were used for immunoblot analysis. Specific antibodies used for the study were: antibodies against cleaved capase-3 (1/1000 dilution, 9664 S), Akt (1/1000 dilution, 9272 S), phospho-Akt (Ser473) (1/1000 dilution, 4060 S), and phospho-myosin light chain 2 (Ser19) (1/1000 dilution, 3675 S) were purchased from Cell Signaling Technology (Danvers, MA, USA); antibodies against p-LIMK-1/2 (Thr508/505) (1/200 dilution, Sc-28409-R), p-Cofilin (mSer 3) (1/200 dilution, Sc-21867-R), and beta Actin (1/200 dilution, Sc-47778) were obtained from Santa Cruz Biotechnology (Santa Cruz, CA, USA); antibodies against Myo9b (1/1000 dilution, 12432-1-AP), DYKDDDDK tag (1/1000 dilution, 20543-1-AP), and PTEN (1/1000 dilution, 22034-1-AP) were ordered from Proteintech (Wuhan, China). The antibody against phospho-PTEN (Ser380) (1/1000 dilution, AP0930) was obtained from Abclonal (Wuhan, China).

### Adoptive transfer model

naive T cells were isolated from splenocytes of 5-week-old BDC2.5 NOD mice with a MojoSort Mouse CD4 naive T-cell Isolation Kit (480040; Biolegend, San Diego, CA, USA) according to the manufacturer's instructions. Female NOD-*scid* mice and ALR *Myo9b* KI NOD-*scid* mice were injected with naive T cells ($2 \times 10^5$ cells per mouse) intravenously. All recipients were monitored for diabetes by measuring blood glucose levels every day, and some of them were euthanized by $CO_2$ inhalation at day 12 following transfer for histological and flow cytometry analysis.

### DC-T-cell co-culture

Splenic naive T cells were enriched from BDC2.5 mice as described earlier and labeled with CFSE. LPS-activated BMDCs were loaded with BDC2.5 mimotope 1040-31 (RP20518; GenScript Biotech, NJ, USA) for 2 h and were then cultured with BDC2.5 naive T cells at a ratio of 1:5. Three days later, T-cell proliferation was determined by flow cytometry as the dilution of CFSE. For the DC-mediated T-cell differentiation

assay, cells were cultured under non-polarized condition. After 3 days of co-culture, cells were harvested for flow cytometry analysis as above.

### In vivo T-cell proliferation assay

naive CD4[+] T cells purified from the spleen of BDC2.5 mice were labeled with CFSE and transferred intravenously into the recipient NOD mice ($1 \times 10^6$ cells/mouse). After 1 day, mice were immunized intravenously with BDC2.5 mimotope-pulsed WT, KO or KI DCs ($3 \times 10^5$ cells/mouse). Three days later, the proliferation of transferred CD4[+] T cells was analyzed by flow cytometry as reflected by dilution of CFSE-fluorescence intensity.

### Glucose uptake assay and seahorse metabolic analysis

BMDCs were starved in glucose-free medium for 2 h, and then incubated with 2-NBDG (N13195; Invitrogen, Carlsbad, CA) at a final concentration of 100 μM for 10 min at 37 °C. The cells were then washed twice with cold PBS and subsequently subjected to flow cytometry analysis.

The extracellular acidification rate (ECAR) and oxygen-consumption rate (OCR) were measured using a Seahorse XFe24 analyzer (Agilent Technologies, Santa Clara, CA, USA). Briefly, BMDCs were seeded in an XF24 culture microplate and allowed to adhere overnight. After LPS stimulation for 24 h, cells were washed with the XF assay medium and pre-equilibrated in a $CO_2$-free incubator for 1 h. EACR was measured under basal condition and after the sequential injection of 10 mM glucose, 1 μM oligomycin, and 50 mM 2-DG (all from Sigma-Aldrich, St. Louis, MO). OCR was measured in the presence of 1.5 μM oligomycin, 1.5 μM FCCP, and 0.5 μM antimycin A/rotenone. All tests were performed in triplicate wells per condition, and ECAR and OCR values were normalized to the protein content. Data were analyzed using the XFe Wave software (Agilent Technologies, Santa Clara, CA, USA).

### Small GTPase RhoA activity assay

The amount of GTP-bound RhoA was evaluated with an Active Rho Pull-Down and Detection Kit (16116; Thermo Fisher, South San Francisco, CA, USA) according to the manufacturer's instructions. Briefly, the cell lysate was incubated at 4 °C for 1 h with a GST fusion protein containing the Rho-binding domain of Rhotekin (GST-Rhotekin-RBD) and glutathione agarose resin. After washing, the resin-bound proteins were eluted, separated by 12% SDS-PAGE and probed with anti-RhoA antibody (1/1000 dilution, 2117 S; Cell Signaling Technology, Danvers, MA, USA). The total cell lysate was also analyzed by immunoblotting with an anti-RhoA antibody to calculate the fractional ratio of GTP-bound RhoA.

### RNA-sequencing and bioinformatic analysis

Total RNA was extracted and used to prepare the libraries with a TruSeq Stranded Total RNA Library Prep Kit (Illumina). Libraries were sequenced on an Illumina HiSeq 2500 instrument. After trimming of adapter sequences and removal of low-quality bases, clean reads were mapped to the mm10 reference genome and the fragments per kilobase of exon per million fragments mapped (FPKM) was calculated using the Bowtie2 software. Differential expression analysis was performed using DESeq2 package (Bioconductor software). The differentially expressed genes (DEGs, FDR < 0.05) were used for subsequent Kyoto Encyclopedia of Genes and Genomes (KEGG) analysis and to generate heatmaps.

### Human samples

The study consisted of 1298 unrelated T1D patients and 2936 healthy controls recruited from the Second Xiangya Hospital of Central South University, the Third Affiliated Hospital of Sun Yat-sen University, and Tongji Hospital. T1D was diagnosed according to the World Health Organization (WHO) criteria, and only patients with the presence of at least one autoantibody against islet (IAA, ZnT8, or GADA) were included as T1D cases. Healthy controls, who did not have overt auto-immune diseases or diabetes mellitus or any other chronic diseases, were enrolled from the same geographical regions. Clinical characteristics of the study population were provided in Supplementary Table 4. PBMCs from 3 T1D patients carrying the *MYO9B* variant and 3 T1D non-carriers with comparative age and disease state were isolated for analysis of MoDCs. Informed consent was obtained from all of the individuals included. The study was performed in accordance with the Declaration of Helsinki and was approved by the Institutional Review Board (IRB) of Tongji Hospital (TJ-IRB20160602).

### Targeted sequencing

Genomic DNA was extracted from human whole blood using a TIA-Namp Blood DNA Kit (DP348; TIANGEN, Beijing, China) following the manufacturer's instructions. Libraries were then individually constructed using genomic DNA and subjected to targeted sequencing (BGI Genomics, Shenzhen, China) of the genomic region covering the *MYO9B* gene. Raw data of sequencing were initially processed, aligned to the human reference genome (hg19/GRCh37) to analyze the coverage and conduct the variant callings.

### Genotyping

Genotyping for the variants Rs764932023, Rs766200985, and Rs776331004 was performed using the Taqman 7900HT Sequence Detection System. The reaction system consisted of 10 ng DNA, Taq-Man universal PCR master mix (Applied Biosystems, CA, USA), forward and reverse primers, and FAM and VIC labeled probes designed by the Tsingke Biotechnology Co., Ltd (Beijing, China). Sequence Detection Systems 2.1 software was used to analyze the results. All DNA samples for controls and cases were run in the same batches, and randomly selected samples measured in duplicate were in complete concordance. The sequences of primers and probes are listed in Supplementary Table 5.

### Adenoviral transduction

The adenoviruses carrying the human *MYO9B* gene (*MYO9B*[WT]) or variant Rs764932023 (*MYO9B*[R133Q]) were packaged by the DesignGene Biotechnology (Shanghai, China). BMDCs from KO mice were plated in 24-well cell culture plates at a density of $4 \times 10^5$ cells/ml and cultured overnight, then transduced with corresponding viruses at 200 MOI. The culture medium was changed 24 h later, and the cells were further cultured for subsequent experiments.

### Statistical analysis

Survival curves were computed with the Kaplan-Meier method and were compared with the log-rank test. The difference in insulitis severity was determined using the $\chi^2$ test. The differences for genotype frequencies between cases and controls were assessed using the Fisher's exact test. Other data were presented as mean ± SEM, and their comparisons were conducted by the two-sided Student's $t$ test or one-way ANOVA where appropriate. Statistical analyses of the data were undertaken with the GraphPad Prism 5 software (GraphPad Software Inc., San Diego, CA, USA). All in vitro studies were conducted with at least three independent replications. For all statistical comparisons, $p < 0.05$ was considered with statistical significance.

### Reporting summary

Further information on research design is available in the Nature Portfolio Reporting Summary linked to this article.

## Data availability

The data that support the findings of this study are provided within the manuscript and Supplementary Information file. The RNA-seq data have

been deposited in the NCBI public repository Sequence Read Archive under accession code SRP400184. The metabolomics data have been deposited in the MetaboLights under accession code MTBLS8150. Mouse mm10 reference genome used in this study is available from https://www.ncbi.nlm.nih.gov/assembly/GCF_000001635.26. Human hg19 reference genome used in this study is available from https://www.ncbi.nlm.nih.gov/assembly/GCF_000001405.25. Unique reagents used in this article are available from the corresponding authors upon reasonable request. The raw numbers for charts and graphs are available in the Source Data file whenever possible. Source data are provided with this paper.

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

## Acknowledgements

We are grateful to Siqian Liu and Danni Song from the Center for Biomedical Research, Tongji Hospital, Tongji Medical College, Huazhong University of Science and Technology for their help in animal experiments. Our study was supported by the National Key R&D Program of China (2022YFA0806101 to C.-Y.W.) from the Ministry of Science and Technology of China, the National Natural Science Foundation of China (82130023 and 81920108009 to C-Y.W., 82100892 to J.Z., 82200923 to F.S., 82270893 to Q.G., 82270885 to Q.L.Y., 82200926 to T.T.Y.,

82070808 to S.Z., 81873656 to F.X., 82100823 to F.X.W., and 82100931 to H.Z.), the Hubei Health Committee Program Project (WJ2021ZH0002 to C.-Y.W. and 2022CFB739 to L.M.C.), the Postdoctoral Science Foundation of China (54000-0106540081 to F.X.W. and 54000-0106540080 to F.S.), the Integrated Innovative Team for Major Human Diseases Program of Tongji Medical College, Huazhong University of Science and Technology, and the Innovative Funding for Translational Research from Tongji Hospital.

## Author contributions

J.Z., Y.Z., and L.M.C. were responsible for conducting all of the experiments and data analyses and preparing the manuscript. F.S., Q.Q.X., Q.Z., Y.W., and N.W. were involved in histological analysis, western blot, real-time PCR, and animal experiments. X.L. provided help for seahorse metabolic analysis. Y.L., S.Z., F.X., and P.Y. jointly performed some of the experiments. S.W.L., T.Y., J.P.W., D.L.E., J.H.Y., and Z.G.Z. were involved in the study design and review of the manuscript, and C.-Y.W. contributed to the study design and manuscript preparation. All authors were involved in drafting the article or revising it critically, and all authors gave their approval for the final paper to be published.

## Competing interests

The authors declare no competing interests.
