## [Peer Review File · Nature Communications]

REVIEWER COMMENTS

Reviewer #1 (Immune metabolism, mouse T1D model) (Remarks to the Author):

The manuscript of Chen L, Zhang J, Zou Y et al. reports that a deletion in the gene *Myo9b* confers resistance against type I diabetes to the ALR/Lt mouse strain, as compared to NOD mice. Upon generation of genetically-modified mice, and mostly using in vitro-generated bone marrow-derived dendritic cells, the Authors suggest that the defective functionality of ALR *Myo9b*, as well as deficiency of the *Myo9b* gene itself, affects the motility, maturation, and metabolism of DCs. Defective DC functions ultimately impinge on the CD4⁺ T cell response. The mouse data are corroborated by the finding that a small fraction of type I diabetes patients (0.54%), as compared to healthy donors, carries a variant in the *Myo9b* gene driving enhanced DC activation and glycolytic metabolism.

The manuscript is synthetic and well written, and the topic of research is of high interest. I liked the approach of comparing resistant vs susceptible mouse strains to identify genetic traits that could explain this difference.

However, I have concerns about the relevance of the experimental approaches used in the manuscript. Similarly, I think some of the results presented throughout the manuscript do not support the conclusions of the Authors.

Here are five points of major criticism that may help to improve/increase the relevance of the manuscript.

1. Most of the experiments rely on the use of bone marrow-derived dendritic cells (BMDCs). I am aware that this is a well validated model, but considering the fact that the Authors are using mice, some of the key results of the manuscript should be validated in ex vivo isolated dendritic cells. I understand that DCs can be isolated only in relatively small amounts from murine tissues, however flow cytometry-based technologies, as well as low input transcriptomic analysis may overcome these hurdles. DC subset distribution, numbers, maturation profiles, cellular metabolism should be assessed by means of flow cytometry, using cell suspensions isolated from the pancreas, as well as pancreatic lymph nodes/mesenteric lymph nodes.
2. The claims following the immune-related results comparing the effect of ALR *Myo9b*, deletion of *Myo9b*, as compared to WT NOD mice are partly misleading. For instance, Figure 2I and 2J show a delay in the onset of insulinitis in KO and KI mice, rather than showing protection from T1D. Tissue infiltration data showed in supplementary figure 2 do not show visible differences, at least to me, in liver, kidney and heart (perhaps flow cytometry could be easier to quantify). I am left wondering if the reduction of IFN γ -producing CD4⁺ T cells from 6% to 3% (Figure 3C) can justify the delay in onset of T1D. Wouldn't the Authors expect a higher fraction of T cells producing IFN γ during T1D inflammation in pancreas-draining LN? The slight increase in Treg numbers upon *Myo9b* KO and ALR *Myo9b* KI should be corroborated by assessing their suppressive function.
3. The whole paper revolves around the role of *Myo9b* in DC function in T1D. Coming back to my concerns raised in 1., the Authors should really assess the role of *Myo9b* in ex vivo isolated DCs. Also, MHC I, CD80 and CD86 are unaltered. What happens to CD8⁺ T cells during the process of cross-presentation? I would suggest that the role of *Myo9b* should be assessed in different DC subsets, for instance in cDC1 and cDC2. Figure 5I, J and K provide hints regarding the ability of BMDCs to prime CD4⁺ T cells, rather than suggesting anything about their ability to shape their diversification. By adding exogenous cytokines, the intrinsic ability of DCs in driving T cell diversification is difficult to quantify.
4. In Figure 7, the Authors claim that use of the Akt agonist SC79 reverts the effects on DC metabolism and function of *Myo9b* deletion and ALR *Myo9b* KI. All the experiments with the agonist must be performed using also the untreated control (WT, KO and KI without SC79).

5. The Authors have access to a generous cohort of T1D patients. Thanks to its extent, the Authors managed to identify that the variant rs764932023 in MYO9B is associated to T1D, despite being identified in a very small fraction of T1D patients. Considering the extent of the cohort, instead of using mouse Myo9b-deficient BMDC transduced with a vector encoding the variant of MYO9B, couldn't the Authors use monocyte-derived DCs from T1D patients carrying the MYO9B variants? This would confer the paper a much higher clinical relevance. Or perhaps a flow cytometry-based immunological analysis of blood DCs in T1D patients vs healthy controls could already be very telling.

Additional concerns/suggestions:

1. Myo9b has a paralogue, Myo9a. What happens to its expression in ALR mice? Does it compensate for Myo9b?
2. Which cell types express Myo9b? Is its expression DC-restricted? Also, Myo9b is important to control cell shape? What happens to the shape of DCs isolated from the pancreatic islets of KO, KI vs WT mice?
3. The colon infiltrate shown in Supp. Figure 2 looks strange. I would expect infiltration of lymphocytes in the lamina propria of the colon, rather than having such a localized infiltrate. Flow cytometry could help you better quantify the infiltrate.
4. The axes of the dot plots in Figures 3B and 3G are likely mislabelled. Please, double check them.
5. Why didn't the Authors generate a DC-specific KI, rather than a KO?
6. The statement that ALR Myo9b protect against T1D is a bit too strong. This variant of Myo9b really delays the onset of T1D. Also, the CD11c-Cre is not targeting exclusively DCs. Also some CD4+ T cells may express CD11c upon activation. I am not suggesting to generate a new model, however corroborating the current results with an in depth analysis of Myo9b role in DCs in vivo could help clarify this doubt.
7. The Authors used a model relying on skin painting with FITC to assess DC recruitment to the LN. Could the Authors assess the recruitment of DCs in the pancreatic LN of T1D mice instead? It would add relevance to their statements.
8. Please clarify how many times the experiments have been performed, in each figure legend. I often did not find these details and I was sometimes confused whether the indicated n was referring to the number of mice used in a single experiment, or the number of experiments.
9. In Figure 5H the flow dot plots are mismatched with the histograms. Please double check this issue.
10. Figure 6E show that KO and KI BMDC have a difference in their reliance to the electron transport chain, besides than in their glycolytic activity, as compared to WT. This effect should not be overlooked, also considering the KEGG pathway analysis in 6B. What does the OCR data show for the different DC genotypes?
11. Metabolomic analysis of the amounts of glycolytic intermediates in WT, KO and KI BMDCs should be used to corroborate a role of the altered NAD⁺/NADH ratio in glycolysis progression.
12. 2-NBDG has been claimed to be a poor readout for glucose uptake (see the work published from the lab of Doreen Cantrell). Could you please use the kynurenine uptake assay (details published from the same lab) as a more reliable proxy of glucose uptake by DCs?

Matteo Villa, PhD
Medical University of Graz
Austria

Reviewer #2 (Dendritic cell biology, T cell activation) (Remarks to the Author):

By comparing the type 1 diabetes (T1D)-prone mouse strain NOD/ShiLtJ to its T1D-resistant congenic strain ALR/Lt, Chen et al. found that DCs from ALR mice were impaired in terms of migration and T

cell priming. They performed genomic sequencing, which revealed a variant in the myosin IXb (Myo9b) gene in ALR mice. Subsequently, they generated NOD mice with ALR Myo9b knock-in, which were much less prone to developing spontaneous T1D. Moreover, they also developed a DC-specific Myo9b knockout, which were even less prone to spontaneous T1D. They found that Myo9b was involved in cellular glycolysis by RhoA/PTEN-mediated Akt inactivation, which in turn resulted in impaired phenotypic and functional maturation of DCs, including impaired migration, low MHC-II, low pro-inflammatory cytokines, and low T cell priming in vitro and in vivo. Lastly, the authors performed targeted high-throughput genomic sequencing in T1D patients, identifying a rare MYO9B R133Q mutation that was associated with a higher risk of T1D.

This is a well presented and very nice translational study, starting from an observation in mice and going all the way of explaining the phenotype and searching its correlate in patients. Below are some suggestions on how this manuscript could be improved.

- 1) In Fig. 3, does the increase in Foxp3+ Treg cell percentages also translate to increased cell counts? Is an increase in Foxp3+ Treg cell percentages (and counts) also observed in the thymus? The authors should provide these data.
- 2) In Fig. 5, why do KO and KI DCs express less MHC-II? Are IL-4 receptors or IFN- γ receptors altered on these DCs in comparison to their WT counterparts?
- 3) In Fig. 2G, it would be helpful if the authors could comment (without having to provide the data) on how the spontaneous development of T1D would look like in ALR mice.

Reviewer #3 (T1D in human, autoimmune disease) (Remarks to the Author):

The study by Chen et al. identifies Myo9b as a potential genetic susceptibility factor in the NOD mouse model, which is the most used animal model of human type 1 diabetes. The researchers first identified a deletion within the Myo9b gene as a potential explanation for resistance to autoimmune diabetes in the ALR mouse strain compared to the NOD strain. Through elaborate knock-in and knock-out mouse models they then proceed to establish that Myo9b-deficiency in fact prevents autoimmune diabetes in the NOD model, and this is likely caused by dysfunctional motility and T-cell activating capacity of dendritic cells. Moreover, they demonstrate that in Myo9b-deficiency the dysfunction of DCs is mechanistically caused by alterations in glycolysis.

Finally, the researchers aim to find genetic polymorphisms within MYO9B that would be linked with type 1 diabetes risk in humans. Through targeted sequencing and Taqman genotyping they discover a rare SNP that appears to be associated with T1D risk. They further demonstrate that this SNP appears to increase MYO9B expression in DCs and increase their T-cell activating capacity in vitro.

The manuscript is clearly written. The mouse experiments performed in the study are well-planned, meticulously performed and technically sound, and convincingly illustrate that Myo9b affects DC functionality and defects in it can increase the risk of autoimmunity in the NOD mouse model. This potentially advances our understanding of the mechanisms of autoimmunity in general as well.

The translational advance of this study towards human type 1 diabetes remains, however, unclear,, since the identified SNP is extremely rare in humans and could therefore explain only extremely rare cases of human type 1 diabetes.

Major comments:

- 1) The MYO9B SNP rs764932023 identified to be associated with T1D risk is and extremely rare variant: MAF < 0.2% in Asian populations and virtually absent in European/American/African populations. This diminishes the translational potential of the study to human T1D, as this variant

cannot be an important driver of T1D. Since more common SNPs have also not been associated with T1D risk in GWAS studies, it remains unclear whether MYO9B itself a major contributor to T1D pathogenesis.

However, this study provides a proof-of-principle that defects in DC function (caused by Myo9b here) can be potential drivers for autoimmunity. Since the effect of common T1D-associated SNPs on DC function is an understudied area, this concept could be mentioned in more detail in Discussion, i.e. that although this study cannot establish a direct role for MYO9B in human type 1 diabetes, it could point to DC function as a potential research area for investigating the impact of genetic variation on T1D risk.

2) Type 1 diabetes is a human disease, and the NOD mouse is a model that. However, it is incorrect to say that "T1D develops in NOD mice", the disease phenotype in mice should rather be called "autoimmune diabetes". Please clarify this throughout the manuscript (for example rows 135, 452, 596)

3) Although it is likely that introduction of Myo9b(ALR/ALR) to the NOD background protects from autoimmune diabetes due to its effect on immune cells this is not firmly established, since the variant was not specifically introduced to immune cells. Theoretically, the Myo9b(ALR/ALR) mutation could also affect other cells in the mouse, for example by increasing beta cell resistance to apoptosis. This should be addressed in the manuscript.

Minor comments:

Row 119: For clarity: "(Idd2) locus is unique..."

Row 124: Myo9b could initially be described a bit more comprehensively, e.g. "Myosin IXb (Myo9b) is a Rho GTPase-activating protein (RhoGAP), which functions as an actin-based molecular motor..."

Row 378: Define NOD and ALR mice better, e.g. "autoimmune diabetes-prone NOD and diabetes-resistant ALR mice"

385: Clarify "function of DCs to promote T cell proliferation in vitro". Similarly, I would emphasize that the experiments were done in vitro in row 482 (In response to LPS stimulation in vitro) and row 492 (promote T cell proliferation in vitro)

Row 391: "Sequence great shift" is not a commonly used expression. Please clarify the term here and in Fig. 1E

Row 427: No statistics shown for data in Fig. S2. to support the statement.

Rows 443-445: Please clarify how Th1 and Th17 were defined in this model, i.e. "lower proportion of IFN-g producing (Th1) CD4+ T cells" and "regarding IL-17A-producing (Th17) CD4+ T cells". Also Tregs could be defined more accurately: "proportion of FOXP3+ CD4+ T cells (Treg)"

Row 447: To improve clarity, I would suggest to state: "The levels of proinflammatory cytokines IL-1b, TNF-a and IFN-g were lower whereas the level of the anti-inflammatory cytokine TGF-b was higher..."

Row 496 and Fig. 5H: The word immunization can be confusing, since it usually means injecting antigen to the animal. I would suggest to rephrase to "Adoptive transfer of DCs loaded in vitro..."

Row 530: "It has been reported" would be more correct here.

Row 541: To improve clarity, rephrase to: "differences observed between ALR and NOD mice (Fig. 1J)"

Row 545: "basal glycolysis AND maximal glycolytic capacity"

Row 567: To improve clarity, please emphasize the the variant is "rare"

Rows 578-588 and Fig. 8F-K: Please emphasize that mouse DCs from Myo9b KO mice were used in these experiments. Did the transfection with MYO9B(R133Q) increase My09b levels in mouse DCs similar to human moDCs or not?

Row 587-588: Here the data is overinterpreted. Rephrase to e.g. "variant correlates with increased MYO9B expression and improved T-cell activating capacity in DCs, potentially giving a mechanistical explanation for its association with increased T1D risk"

Row 607. I would rephrase this to "immunocytes other than T cells likely played a major role..." Although unlikely, Myo9bALR/ALR could lead to diabetes resistance due to changes in other cell types, for example increased beta cell resistance to apoptosis.

Fig. 2J: Is the severity in insulitis statistically significant between WT, KO and KI mice? This should be clarified. The differences especially at weeks 12 and 26 look minor.

Fig.3b and 3g: Are the labels for CD44hiCD62Llo vs. CD44loCD62Lhi reversed?

Fig. 8C: Please provide CI for the OR.

Reviewer #1 (Immune metabolism, mouse T1D model) (Remarks to the Author):

The manuscript of Chen L, Zhang J, Zou Y et al. reports that a deletion in the gene Myo9b confers resistance against type I diabetes to the ALR/Lt mouse strain, as compared to NOD mice. Upon generation of genetically-modified mice, and mostly using in vitro-generated bone marrow-derived dendritic cells, the Authors suggest that the defective functionality of ALR Myo9b, as well as deficiency of the Myo9b gene itself, affects the motility, maturation, and metabolism of DCs. Defective DC functions ultimately impinge on the CD4+ T cell response. The mouse data are corroborated by the finding that a small fraction of type I diabetes patients (0.54%), as compared to healthy donors, carries a variant in the Myo9b gene driving enhanced DC activation and glycolytic metabolism.

The manuscript is synthetic and well written, and the topic of research is of high interest. I liked the approach of comparing resistant vs susceptible mouse strains to identify genetic traits that could explain this difference. However, I have concerns about the relevance of the experimental approaches used in the manuscript. Similarly, I think some of the results presented throughout the manuscript do not support the conclusions of the Authors.

Here are five points of major criticism that may help to improve/increase the relevance of the manuscript.

1. Most of the experiments rely on the use of bone marrow-derived dendritic cells (BMDCs). I am aware that this is a well validated model, but considering the fact that the Authors are using mice, some of the key results of the manuscript should be validated in ex vivo isolated dendritic cells. I understand that DCs can be isolated only in relatively small amounts from murine tissues, however flow cytometry-based technologies, as well as low input transcriptomic analysis may overcome these hurdles. DC subset distribution, numbers, maturation profiles, cellular metabolism should be assessed by means of flow cytometry, using cell suspensions isolated from the pancreas, as well as pancreatic lymph nodes/mesenteric lymph nodes.

Response: We highly appreciate this constructive suggestion, which should improve our manuscript a lot. In order to address this question, we first assessed DC numbers, subset distribution, and maturation profiles using cell suspensions isolated from the pancreatic LNs and pancreas. Flow cytometry analysis revealed that both ALR *Myo9b* KI and *Myo9b* deficiency resulted in decreased DC numbers in the PLNs and pancreas (Fig. 3A and B), but without a perceptible impact on cDC1/cDC2 distribution (Fig. 3E and F). Moreover, ALR *Myo9b* KI and *Myo9b*^{-/-} DCs showed decreased expression of MHC class II in the PLNs and pancreas (Fig. 3C and D), while CD80, CD86 and MHC I expression did not show a significant difference (Fig. S3A and B). Next, we examined cellular metabolism by Seahorse. Given that the number of DCs are extremely low in the PLNs and pancreas, and the isolation procedures are complex, which require lengthy enzymatic digestion steps that may affect their metabolic phenotype, we, therefore, used splenic DCs for the study. ECAR analysis indicated an ablated glycolytic metabolism in KO and KI splenic DCs (Fig. 6I and J), all of which were consistent with our BMDC results.

2. The claims following the immune-related results comparing the effect of ALR *Myo9b*, deletion of *Myo9b*, as compared to WT NOD mice are partly misleading. For instance, Figure 2I and 2J show a delay in the onset of insulinitis in KO and KI mice, rather than showing protection from T1D. Tissue infiltration data showed in supplementary figure 2 do not show visible differences, at least to me, in liver, kidney and heart (perhaps flow cytometry could be easier to quantify). I am left wondering if the reduction of IFN γ -producing CD4⁺ T cells from 6% to 3% (Figure 3C) can justify the delay in onset of T1D. Wouldn't the Authors expect an higher fraction of T cells producing IFN γ during T1D inflammation in pancreas-draining LN? The slight increase in Treg numbers upon *Myo9b* KO and ALR *Myo9b* KI should be corroborated by assessing their suppressive function.

Response: Thanks for raising these questions. (1) We have changed the statement of “protect...from/against T1D” to “prevent the development and progression of autoimmune diabetes” in the revised manuscript. (2) Indeed, lymphocytic infiltrates were not evident in the kidney, liver, and heart at the time point we examined, just as we described in the main text. Following your advice, we further performed flow cytometry of tissues to quantify the infiltrates, and included the data into Fig. S2B-G. (3) The results of Th1 cell proportions shown in Fig. 4D (original Fig. 3C) were the representative results from more than one batch of mice (a total of 12 mice), which

were comparable in different batches. Additionally, to exclude the potential bias brought by reagent, we re-conducted the experiment using another fluorescence-labeled IFN- γ antibody, but still did not detect a higher fraction of Th1 cells. Similar and even lower percentages of Th1 cells were also demonstrated in other literature (1-3), possibly due to different breeding environment. Therefore, we believe that the results reflect the real situation in prediabetic NOD mice (10- to 12-week-old). To further address this question, we examined the proportion of IFN- γ producing Th1 cells in the PLNs in 16-week-old mice. Much higher proportion of Th1 cells was observed (Fig. S4A), and the difference between three groups of mice was consistent with that of prediabetic mice. (4) As requested, we conducted Treg suppressive assay and found that Treg cells enriched from KO and KI mice were more efficient in suppressing the proliferation of conventional T cells compared with their WT counterparts (Fig. 4O).

3. The whole paper revolves around the role of Myo9b in DC function in T1D. Coming back to my concerns raised in 1., the Authors should really assess the role of Myo9b in ex vivo isolated DCs. Also, MHC I, CD80 and CD86 are unaltered. What happens to CD8+ T cells during the process of cross-presentation? I would suggest that the role of Myo9b should be assessed in different DC subsets, for instance in cDC1 and cDC2. Figure 5I, J and K provide hints regarding the ability of BMDCs to prime CD4+ T cells, rather than suggesting anything about their ability to shape their diversification. By adding exogenous cytokines, the intrinsic ability of DCs in driving T cell diversification is difficult to quantify.

Response: (1) As addressed earlier, we have assessed the role of Myo9b in *ex vivo* isolated DCs. (2) To answer the above question, we examined CD8⁺ T cell status in the spleen and PLNs by flow cytometry analysis. The KO and KI mice exhibited lower proportions of CD8⁺CD44^{hi}CD62L^{lo} effector T cells (Fig. 4C and J) and IFN- γ ⁺ cytotoxic CD8⁺ T cells (Fig. 4E and L) in the spleen and PLNs. We then assessed the role of Myo9b in cDC1 and cDC2. In line with total DCs, ALR *Myo9b* KI and *Myo9b*^{-/-} cDC1/cDC2 showed decreased expression of MHC class II, but the expression levels of MHC class I and co-stimulatory molecules CD80 and CD86 were almost unchanged (Fig. S3C-F). Since the activation of CD8⁺ T cells also depends on CD4⁺ T cells, which are the major culprits due to their enormous impact on the amplification of autoimmunity in T1D setting, and cDC2 is the predominant subpopulation in the PLNs and pancreas, we therefore in this report selectively focused on the impact of altered DC maturation on CD4⁺ T cells. In our future studies we would tackle the effect of ALR *Myo9b* KI and *Myo9b*^{-/-} DCs on CD8⁺ T cells during the process of cross-presentation

and its association with DC classification and contribution. Relevant description and discussion have been added into the revised manuscript. (3) Thanks for your interpretation of our original Fig. 5I-K, we thus did DC-T cell co-culture again without adding exogenous cytokines. Consistent with less proinflammatory cytokine secretion, ALR *Myo9b* KI and *Myo9b* deficiency DCs reduced their ability to prime Th1 cells (Figure 3O), and similar effect was observed on Th17 cells to a less extent (Figure 3P), but markedly upregulated their capability to induce Treg cells (Figure 3Q).

4. In Figure 7, the Authors claim that use of the Akt agonist SC79 reverts the effects on DC metabolism and function of *Myo9b* deletion and ALR *Myo9b* KI. All the experiments with the agonist must be performed using also the untreated control (WT, KO and KI without SC79).

Response: We appreciate very much for pointing out this defect. Therefore, we re-conducted all of these experiments in the presence/absence of SC79. The corresponding data were updated in Fig. 7.

5. The Authors have access to a generous cohort of T1D patients. Thanks to its extent, the Authors managed to identify that the variant rs764932023 in MYO9B is associated to T1D, despite being identified in a very small fraction of T1D patients. Considering the extent of the cohort, instead of using mouse *Myo9b*-deficient BMDC transduced with a vector encoding the variant of MYO9B, couldn't the Authors use monocyte-derived DCs from T1D patients carrying the MYO9B variants? This would confer the paper a much higher clinical relevance. Or perhaps a flow cytometry-based immunological analysis of blood DCs in T1D patients vs healthy controls could already be very telling.

Response: At the very beginning, we planned to use MoDC from subjects carrying the *Myo9B* variant to conduct experiments in this part. However, since the human samples were collected at multiple centers over a long period of time, many of these subjects were unable to get new contact information, while the others did not agree to donate enough peripheral blood for subsequent studies. As a compromised approach, we transduced *Myo9b*^{-/-} DCs with *MYO9B*^{R133Q} adenovirus to characterize the impact of the R133Q polymorphism on DC activities. To further address the above question, we tried our best for 2 months to get in touch with three of the T1D patients who carry the *MYO9B*^{R133Q} polymorphism with comparative age and disease state, and repeated the experiments using moDCs. Indeed, the patients carrying the *MYO9B*^{R133Q}

polymorphism manifested higher MHC II expression coupled with increased inflammatory cytokine expression, higher capacity to prime CD4 T cells, and higher glycolytic metabolism (Fig. 8F-J).

Additional concerns/suggestions:

1. Myo9b has a paralogue, Myo9a. What happens to its expression in ALR mice? Does it compensate for Myo9b?

Response: Myo9a expression was comparable between NOD and ALR mice, and the data was added into Fig. S1B.

2. Which cell types express Myo9b? Is its expression DC-restricted? Also, Myo9b is important to control cell shape? What happens to the shape of DCs isolated from the pancreatic islets of KO, KI vs WT mice?

Response: In the Introduction part, we had described that Myo9b is abundantly expressed in various cell types of the immune system and contributes to the regulation of their morphology and motility. To further answer this question, we isolated DCs from the pancreatic islets of WT, KO, and KI mice, cultured overnight, and carried out phalloidin staining to assess morphological changes. In line with previous studies, Myo9b^{-/-} DCs lost their dendritic morphology and were spherical with very short dendrites as compared to that of WT DCs. Similar morphology was also noted in the KI DCs, but the change was less significant (shown in Fig. 3G).

3. The colon infiltrate shown in Supp. Figure 2 looks strange. I would expect infiltration of lymphocytes in the lamina propria of the colon, rather than having such a localized infiltrate. Flow cytometry could help you better quantify the infiltrate.

Response: Thanks for the suggestion. We have quantified the lymphocytic infiltrates in different tissues by flow cytometry, and the result was used in place of the original one in Fig. S2.

4. The axes of the dot plots in Figures 3B and 3G are likely mislabeled. Please, double check them.

Response: We are sorry for the mistake. We have corrected it in the revised manuscript.

5. Why didn't the Authors generate a DC-specific KI, rather than a KO?

Response: Thanks for raising this question. As the 33-bp deletion of *Myo9b* detected in ALR mice is a germline mutation, whole body KI could better mimic the actual situation. We thus generated a whole body KI mice, and the effect of ALR *Myo9b* KI on DC function could be demonstrated by a series of *in vivo*, *in vitro*, and *ex vivo* experiments.

6. The statement that ALR *Myo9b* protect against T1D is a bit too strong. This variant of *Myo9b* really delays the onset of T1D. Also, the CD11c-Cre is not targeting exclusively DCs. Also some CD4⁺ T cells may express CD11c upon activation. I am not suggesting to generate a new model, however corroborating the current results with an in depth analysis of *Myo9b* role in DCs *in vivo* could help clarify this doubt.

Response: We really appreciate this friendly reminder, and have changed the statement to "...prevent the development and progression of autoimmune diabetes" in the revised manuscript. To exclude the potential effect of CD11c-Cre on CD4⁺ T cells, we detected *Myo9b* level in WT and KO CD4⁺ T cells. Actually, no perceptible difference was observed (shown in Fig. S2A), indicating that CD11c-Cre had little impact on *Myo9b* expression in CD4⁺ T cells. Moreover, we analyzed the role of *Myo9b* in DCs, including their number, subset distribution, and maturation profile *in vivo*, as well as cellular metabolism *ex vivo* as we replied to major Question 1.

7. The Authors used a model relying of skin painting with FITC to assess DC recruitment to the LN. Could the Authors assess the recruitment of DCs in the pancreatic LN of T1D mice instead? It would add relevance to their statements.

Response: Many thanks for raising this question. To address this question, we injected CFSE-labeled WT, KO and KI DCs into NOD recipient mice, and assessed the proportions of CFSE⁺ cells migrated into the pancreatic LNs by flow cytometry. Consistently, ALR *Myo9b* KI and *Myo9b* deficiency decreased DC recruitment toward PLNs, and the resulting data were employed to replace previous data obtained from FITC-induced DC migration assay (Fig. 3L).

8. Please clarify how many times the experiments have been performed, in each figure legend. I often did not find these details and I was sometimes confused whether the indicated n was referring to the number of mice used in a single experiment, or the number of experiments.

Response: We sincerely apologize for the confusion. In all cases, n referred to the number of mice used in a single experiment, and we had summarized how many times the experiments were performed at the end of each figure legend.

9. In Figure 5H the flow dot plots are mismatched with the histograms. Please double check this issue.

Response: The histogram in Fig. 3N (original Fig. 5H) only showed the boxed areas in the flow dot plots, which represented the adoptively transferred CFSE-labeled BDC2.5 CD4⁺ T cells. We are sorry for the misunderstanding, and we have added an arrow between the boxed area and the histogram for clarification.

10. Figure 6E show that KO and KI BMDC have a difference in their reliance to the electron transport chain, besides than in their glycolytic activity, as compared to WT. This effect should not be overlooked, also considering the KEGG pathway analysis in 6B. What does the OCR data show for the different DC genotypes?

Response: In order to address this question, we measured the OCR of WT, KO, and KI DCs. As shown in Fig. S6, we failed to detect a discernable difference in the OCR level between different DC genotypes.

11. Metabolomic analysis of the amounts of glycolytic intermediates in WT, KO and KI BMDCs should be used to corroborate a role of the altered NAD⁺/NADH ratio in glycolysis progression.

Response: Thanks for your advice. NAD⁺/NADH ratio alone may not be enough for the assessment of glycolysis progression, we therefore used targeted metabolomics to monitor global metabolite changes. The results indicated that the amounts of several key glycolytic intermediates were significantly decreased in KO and KI BMDCs, including D-Glucose-6-phosphate, D-Fructose-6-phosphate, D-Glucose-1-phosphate, glyceraldehyde-3-phosphate, dihydroxyacetone-phosphate, cyclic-AMP, and lactate

(Fig. 6H).

12. 2-NBDG has been claimed to be a poor readout for glucose uptake (see the work published from the lab of Doreen Cantrell). Could you please use the kynurenine uptake assay (details published from the same lab) as a more reliable proxy of glucose uptake by DCs?

Response: Based on the suggestion, we read the paper published from the lab of Doreen A. Cantrell carefully. Their data show that there was a discordance between flow cytometric 2-NBDG staining against radiolabeled glucose transport assays in murine T cells. They also found that 2-NBDG uptake into murine T cells was not inhibited by competitive substrates or facilitative glucose transporter inhibitors, nor can 2-NBDG competitively block glucose uptake in T cells (4). However, in Cos1 cells and in MIN6 pancreatic beta cells, the uptake of 2-NBDG was shown to be competitively inhibited by excess D-glucose and blocked in the presence of glucose transporter inhibitor, cytochalasin B (5). In kidney cells 2-NBDG uptake was mediated by a sodium-glucose linked transporter (SGLT), which was also decreased in response to co-exposure to the SGLT substrate, and could be blocked with the SGLT inhibitor, phlorizin (6). Therefore, this reliability of 2-NBDG uptake might be cell-type dependent. As far as we know, there seems to be no evidence that it is not a feasible tool for the assessment of glucose uptake capacity in DCs. Combined with RNA-seq data, metabolomic analysis, and other results, it could illustrate that glycolytic pathway is impaired. Secondly, kynurenine uptake assay established by the lab of Doreen A. Cantrell is more like a method to assess System L amino acid transport instead of glucose (7, 8).

Reviewer #2 (Dendritic cell biology, T cell activation) (Remarks to the Author):

By comparing the type 1 diabetes (T1D)-prone mouse strain NOD/ShiLtJ to its T1D-resistant congenic strain ALR/Lt, Chen et al. found that DCs from ALR mice were impaired in terms of migration and T cell priming. They performed genomic sequencing, which revealed a variant in the myosin IXb (Myo9b) gene in ALR mice. Subsequently, they generated NOD mice with ALR Myo9b knock-in, which were much less prone to developing spontaneous T1D. Moreover, they also developed a DC-specific Myo9b knockout, which were even less prone to spontaneous T1D. They found that Myo9b was involved in cellular glycolysis by RhoA/PTEN-mediated Akt inactivation, which in turn resulted in impaired phenotypic and functional maturation of DCs, including

impaired migration, low MHC-II, low pro-inflammatory cytokines, and low T cell priming in vitro and in vivo. Lastly, the authors performed targeted high-throughput genomic sequencing in T1D patients, identifying a rare MYO9B R133Q mutation that was associated with a higher risk of T1D.

This is a well presented and very nice translational study, starting from an observation in mice and going all the way of explaining the phenotype and searching its correlate in patients. Below are some suggestions on how this manuscript could be improved.

1) In Fig. 3, does the increase in Foxp3⁺ Treg cell percentages also translate to increased cell counts? Is an increase in Foxp3⁺ Treg cell percentages (and counts) also observed in the thymus? The authors should provide these data.

Response: We really appreciate this constructive suggestion. Yes, the increase in Foxp3⁺ Treg cell percentages also translated to increased cell counts in the PLNs and spleen, and the results were added into Fig. S4B and C as requested. However, we did not observe an increase in terms of percentages and counts of Treg cells in the thymus, which was shown in Fig. S4D and E.

2) In Fig. 5, why do KO and KI DCs express less MHC-II? Are IL-4 receptors or IFN- γ receptors altered on these DCs in comparison to their WT counterparts?

Response: After encountering microbial products or inflammatory stimuli, DCs become activated and matured, a process that involves upregulation of MHC II, secretion of proinflammatory cytokines, migration, and effective T cell priming capacity. Growing evidence suggests that maturation of DC is accompanied by, and dependent on a switch from oxidative phosphorylation to aerobic glycolysis (9). Glycolysis is capable of producing ATP more quickly and generating intermediates for de novo synthesis of amino acids and lipids to support the demand of cellular functions and biosynthetic activities during DC maturation (10, 11). In our study, ALR *Myo9b* KI and *Myo9b*^{-/-} DCs were characterized by the decreased commitment to Akt-dependent glycolysis, and addition of Akt agonist, SC79, remarkably augmented the glycolytic metabolism and MHC II expression in ALR *Myo9b* KI and *Myo9b* deficient DCs to a comparative level as that of WT DCs, indicating that attenuated MHC II expression in KO and KI DCs mainly resulted from disrupted glycolytic process.

Indeed, IFN- γ and IL-4 signaling may also affect MHC II expression, so we checked

the expression of their receptors following your suggestion. However, no significant difference was noted between groups, and the data were added into Fig. S3I.

3) In Fig. 2G, it would be helpful if the authors could comment (without having to provide the data) on how the spontaneous development of T1D would look like in ALR mice.

Response: ALR mice do not develop spontaneous T1D, which we have stated in the Abstract, Introduction and Discussion sections.

Reviewer #3 (T1D in human, autoimmune disease) (Remarks to the Author):

The study by Chen et al. identifies Myo9b as a potential genetic susceptibility factor in the NOD mouse model, which is the most used animal model of human type 1 diabetes. The researchers first identified a deletion within the Myo9b gene as a potential explanation for resistance to autoimmune diabetes in the ALR mouse strain compared to the NOD strain. Through elaborate knock-in and knock-out mouse models they then proceed to establish that Myo9b-deficiency in fact prevents autoimmune diabetes in the NOD model, and this is likely caused by dysfunctional motility and T-cell activating capacity of dendritic cells. Moreover, they demonstrate that in Myo9b-deficiency the dysfunction of DCs is mechanistically caused by alterations in glycolysis.

Finally, the researchers aim to find genetic polymorphisms within MYO9B that would be linked with type 1 diabetes risk in humans. Through targeted sequencing and Taqman genotyping they discover a rare SNP that appears to be associated with T1D risk. They further demonstrate that this SNP appears to increase MYO9B expression in DCs and increase their T-cell activating capacity in vitro.

The manuscript is clearly written. The mouse experiments performed in the study are well-planned, meticulously performed and technically sound, and convincingly illustrate that Myo9b affects DC functionality and defects in it can increase the risk of autoimmunity in the NOD mouse model. This potentially advances our understanding of the mechanisms of autoimmunity in general as well.

The translational advance of this study towards human type 1 diabetes remains, however, unclear, since the identified SNP is extremely rare in humans and could therefore explain only extremely rare cases of human type 1 diabetes.

Major comments:

1) The MYO9b SNP rs764932023 identified to be associated with T1D risk is and extremely rare variant: MAF < 0.2% in Asian populations and virtually absent in European/American/African populations. This diminishes the translational potential of the study to human T1D, as this variant cannot be an important driver of T1D. Since more common SNPs have also not been associated with T1D risk in GWAS studies, it remains unclear whether MYO9B itself a major contributor to T1D pathogenesis.

However, this study provides a proof-of-principle that defects in DC function (caused by Myo9b here) can be potential drivers for autoimmunity. Since the effect of common T1D-associated SNPs on DC function is an understudied area, this concept could be mentioned in more detail in Discussion, i.e. that although this study cannot establish a direct role for MYO9B in human type 1 diabetes, it could point to DC function as a potential research area for investigating the impact of genetic variation on T1D risk.

Response: Thank you very much for your support and suggestions. In fact, there is feasible evidence that MYO9B could be a risk factor for autoimmune diseases such as systemic lupus erythematosus (SLE), celiac disease (CD) and rheumatoid arthritis (RA) (12). As a polygenic autoimmune disease, genetic heterogeneity is a common event in type 1 diabetes. For example, PTPN22 has been recognized as a T1D risk gene in Caucasians (13), while the R620W polymorphism is absent in the Asian populations. As requested by this reviewer, we have added more relevant discussions of this issue into the revised manuscript.

2) Type 1 diabetes is a human disease, and the NOD mouse is a model that. However, it is incorrect to say that “T1D develops in NOD mice”, the disease phenotype in mice should rather be called “autoimmune diabetes”. Please clarify this throughout the manuscript (for example rows 135, 452, 596).

Response: We have replaced “T1D” with “autoimmune diabetes” where appropriate.

3) Although it is likely that introduction of Myo9b(ALR/ALR) to the NOD background protects from autoimmune diabetes due to its effect on immune

cells this is not firmly established, since the variant was not specifically introduced to immune cells. Theoretically, the Myo9b(ALR/ALR) mutation could also affect other cells in the mouse, for example by increasing beta cell resistance to apoptosis. This should be addressed in the manuscript.

Response: Based on the suggestion, pancreatic islets were isolated from 5- to 6-week-old WT and KI mice, followed by stimulation with pro-inflammatory cytokine cocktail (IL-1 β + TNF- α + IFN- γ) to induce apoptosis. There was no significant difference in terms of cleaved caspase-3 expression between these two groups of islets, as shown in Fig. S5. As we mentioned earlier, other types of immune cells could be also affected by the KI of ALR Myo9b, while we only studied its impact with focus on DCs in the present report, its implication in other types of immune cells would be investigated in our follow-up studies. We have added more description on this issue in the revised manuscript.

Minor comments:

Row 119: For clarity: "(Idd)22 locus is unique..."

Response: we added the word "locus" for clarification.

Row 124: Myo9b could initially be described a bit more comprehensively, e.g. "Myosin IXb (Myo9b) is a Rho GTPase-activating protein (RhoGAP), which functions as an actin-based molecular motor..."

Response: We comprehensively described Myo9b as suggested when it was first mentioned in the Introduction section.

Row 378: Define NOD and ALR mice better, e.g. "autoimmune diabetes-prone NOD and diabetes-resistant ALR mice"

Response: We defined NOD and ALR mice as "autoimmune diabetes-prone NOD mice" and "autoimmune diabetes-resistant ALR mice" according to your suggestion.

385: Clarify "function of DCs to promote T cell proliferation in vitro". Similarly, I would emphasize that the experiments were done in vitro in row 482 (In response to LPS stimulation in vitro) and row 492 (promote T cell proliferation in vitro)

Response: Following your suggestion, we have clarified that these experiments were performed *in vitro* in the revised manuscript.

Row 391: “Sequence great shift” is not a commonly used expression. Please clarify the term here and in Fig. 1E

Response: We regret for not clarifying enough the term “sequence great shift”, which refers to exonic InDels that are not integral multiples of 3 or more than 12 bp. We added the explanation in the main text and figure legend.

Row 427: No statistics shown for data in Fig. S2. to support the statement.

Response: Thanks for pointing out this defect, which is a similar question raised by Reviewer 1 (minor Question 3). We have quantified the infiltrate using flow cytometry, and included the data into Fig. S2B-G.

Rows 443-445: Please clarify how Th1 and Th17 were defined in this model, i.e. “lower proportion of IFN-g producing (Th1) CD4+ T cells” and “regarding IL-17A-producing (Th17) CD4+ T cells”. Also Tregs could be defined more accurately: “proportion of FOXP3+ CD4+ T cells (Treg)”

Response: We clarified Th1, Th17, and Treg cells as suggested.

Row 447: To improve clarity, I would suggest to state: “The levels of proinflammatory cytokines IL-1b, TNF-a and IFN-g were lower whereas the level of the anti-inflammatory cytokine TGF-b was higher...”

Row 496 and Fig. 5H: The word immunization can be confusing, since it usually means injecting antigen to the animal. I would suggest to rephrase to “Adoptive transfer of DCs loaded in vitro...”

Row 530: “It has been reported” would be more correct here.

Row 541: To improve clarity, rephrase to: “differences observed between ALR and NOD mice (Fig. 1J)”

Row 545: "basal glycolysis AND maximal glycolytic capacity"

Row 567: To improve clarity, please emphasize the variant is "rare"

Response: Many thanks for pointing out these issues. We have changed the text as suggested.

Rows 578-588 and Fig. 8F-K: Please emphasize that mouse DCs from Myo9b KO mice were used in these experiments. Did the transfection with MYO9B(R133Q) increase Myo9b levels in mouse DCs similar to human moDCs or not?

Response: We emphasized that DCs from *Myo9b* KO mice were used in our experiments. Indeed, transduction of *MYO9B*^{R133Q} increased MYO9B levels in mouse DCs similar as human moDCs, and the data are shown in Fig. S9.

Row 587-588: Here the data is overinterpreted. Rephrase to e.g. "variant correlates with increased MYO9B expression and improved T-cell activating capacity in DCs, potentially giving a mechanistical explanation for its association with increased T1D risk"

Row 607. I would rephrase this to "immunocytes other than T cells likely played a major role..." Although unlikely, Myo9bALR/ALR could lead to diabetes resistance due to changes in other cell types, for example increased beta cell resistance to apoptosis.

Response: We have rephrased these two sentences according to the suggestion.

Fig. 2J: Is the severity in insulinitis statistically significant between WT, KO and KI mice? This should be clarified. The differences especially at weeks 12 and 26 look minor.

Response: We have added statistical results for Fig. 2J as pointed out.

Fig.3b and 3g: Are the labels for CD44hiCD62Llo vs. CD44loCD62Lhi reversed?

Response: We are sorry for the mistake. We have corrected it in the revised version.

Fig. 8C: Please provide CI for the OR.

Response: As requested, we have provided 95% CI for the OR in Fig. 8 and Fig. S8.

Once again, we appreciate the enthusiasm from the editorial committee and all reviewers. We have made necessary corrections suggested by you and the reviewers. We hope that the manuscript is now ready for publication in the **Nature Communications**. Should you have additional questions, please kindly inform me.

References

1. M. Lo Conte, M. Antonini Cencicchio, M. Ulaszewska, A. Nobili, I. Cosorich, R. Ferrarese, L. Massimino, A. Andolfo, F. Ungaro, N. Mancini, M. Falcone, A diet enriched in omega-3 PUFA and inulin prevents type 1 diabetes by restoring gut barrier integrity and immune homeostasis in NOD mice. *Front Immunol* **13**, 1089987 (2022).
2. R. Jiang, Y. Qin, Y. Wang, X. Xu, H. Chen, K. Xu, M. Zhang, Dynamic Number and Function of IL-10-Producing Regulatory B Cells in the Immune Microenvironment at Distinct Stages of Type 1 Diabetes. *J Immunol* **208**, 1034–1041 (2022).
3. M. J. Rahman, K. B. Rodrigues, J. A. Quiel, Y. Liu, V. Bhargava, Y. Zhao, C. Hotta-Iwamura, H. Y. Shih, A. W. Lau-Kilby, A. M. Malloy, T. W. Thoner, K. V. Tarbell, Restoration of the type I IFN-IL-1 balance through targeted blockade of PTGER4 inhibits autoimmunity in NOD mice. *JCI Insight* **3**, (2018).
4. L. V. Sinclair, C. Barthelemy, D. A. Cantrell, Single Cell Glucose Uptake Assays: A Cautionary Tale. *Immunometabolism* **2**, e200029 (2020).
5. K. Yamada, M. Nakata, N. Horimoto, M. Saito, H. Matsuoka, N. Inagaki, Measurement of glucose uptake and intracellular calcium concentration in single, living pancreatic beta-cells. *J Biol Chem* **275**, 22278–22283 (2000).
6. A. B. Blodgett, R. K. Kothinti, I. Kamyshko, D. H. Petering, S. Kumar, N. M. Tabatabai, A fluorescence method for measurement of glucose transport in kidney cells. *Diabetes Technol Ther* **13**, 743–751 (2011).
7. L. V. Sinclair, D. Neyens, G. Ramsay, P. M. Taylor, D. A. Cantrell, Single cell analysis of kynurenine and System L amino acid transport in T cells. *Nat Commun* **9**, 1981 (2018).
8. K. M. Grzes, D. E. Sanin, A. M. Kabat, M. A. Stanczak, J. Edwards-Hicks, M. Matsushita, A. Hackl, F. Hassler, K. Knoke, S. Zahalka, M. Villa, D. M. Kofler, R. E. Voll, P. Zigrino, M. Fabri, E. L. Pearce, E. J. Pearce, Plasmacytoid dendritic cell activation is dependent on coordinated expression of distinct amino acid transporters. *Immunity* **54**, 2514–2530 e2517 (2021).
9. T. A. Patente, L. R. Pelgrom, B. Everts, Dendritic cells are what they eat: how their metabolism shapes T helper cell polarization. *Curr Opin Immunol* **58**, 16–23 (2019).
10. C. M. Krawczyk, T. Holowka, J. Sun, J. Blagih, E. Amiel, R. J. DeBerardinis, J. R. Cross, E. Jung, C. B. Thompson, R. G. Jones, E. J. Pearce, Toll-like receptor-induced changes in glycolytic metabolism regulate dendritic cell activation. *Blood* **115**, 4742–4749 (2010).
11. B. Everts, E. Amiel, S. C. Huang, A. M. Smith, C. H. Chang, W. Y. Lam, V. Redmann, T. C. Freitas, J. Blagih, G. J. van der Windt, M. N. Artyomov, R. G. Jones, E. L. Pearce, E. J. Pearce, TLR-driven early glycolytic reprogramming via the kinases TBK1-IRK1ε supports the anabolic demands of dendritic cell activation. *Nat Immunol* **15**, 323–332 (2014).
12. E. Sanchez, B. Z. Alizadeh, G. Valdigem, N. Ortego-Centeno, J. Jimenez-Alonso,

- E. de Ramon, A. Garcia, M. A. Lopez-Nevot, C. Wijmenga, J. Martin, B. P. Koeleman, MYO9B gene polymorphisms are associated with autoimmune diseases in Spanish population. *Hum Immunol* **68**, 610-615 (2007).
13. L. A. Criswell, K. A. Pfeiffer, R. F. Lum, B. Gonzales, J. Novitzke, M. Kern, K. L. Moser, A. B. Begovich, V. E. Carlton, W. Li, A. T. Lee, W. Ortmann, T. W. Behrens, P. K. Gregersen, Analysis of families in the multiple autoimmune disease genetics consortium (MADGC) collection: the PTPN22 620W allele associates with multiple autoimmune phenotypes. *Am J Hum Genet* **76**, 561-571 (2005).

REVIEWERS' COMMENTS

Reviewer #1 (Remarks to the Author):

The Authors have responded well to all of my comments.
I am satisfied with the amended version of the manuscript and figures.

Matteo Villa

Medical University of Graz
Austria

Reviewer #2 (Remarks to the Author):

With their revised manuscript, the authors have satisfactorily addressed my previous comments.

Reviewer #3 (Remarks to the Author):

The authors have satisfactorily addressed my concerns in the revised version of the manuscript.